# Sample-Efficiency in Multi-Batch Reinforcement Learning: The Need for Dimension-Dependent Adaptivity

**Emmeran Johnson & Ciara Pike-Burke**
Department of Mathematics, Imperial College London, United Kingdom
{emmeran.johnson17,c.pike-burke}@imperial.ac.uk

**Patrick Rebeschini**
Department of Statistics, University of Oxford, United Kingdom
patrick.rebeschini@stats.ox.ac.uk

## Abstract

We theoretically explore the relationship between sample-efficiency and adaptivity in reinforcement learning. An algorithm is sample-efficient if it uses a number of queries $n$ to the environment that is polynomial in the dimension $d$ of the problem. Adaptivity refers to the frequency at which queries are sent and feedback is processed to update the querying strategy. To investigate this interplay, we employ a learning framework that allows sending queries in $K$ batches, with feedback being processed and queries updated after each batch. This model encompasses the whole adaptivity spectrum, ranging from non-adaptive 'offline' ($K = 1$) to fully adaptive ($K = n$) scenarios, and regimes in between. For the problems of policy evaluation and best-policy identification under $d$-dimensional linear function approximation, we establish $\Omega(\log \log d)$ lower bounds on the number of batches $K$ required for sample-efficient algorithms with $n = O(poly(d))$ queries. Our results show that just having adaptivity ($K > 1$) does not necessarily guarantee sample-efficiency. Notably, the adaptivity-boundary for sample-efficiency is not between offline reinforcement learning ($K = 1$), where sample-efficiency was known to not be possible, and adaptive settings. Instead, the boundary lies between different regimes of adaptivity and depends on the problem dimension.

## 1 Introduction

Data collection in Reinforcement Learning (RL) usually falls into two main paradigms: online and offline. An online learner interacts with the environment, immediately receiving feedback and adapting its decisions in real-time. In contrast in offline RL, the dataset is collected in a single batch prior to observing any feedback. Many practical applications consider RL algorithms with limited adaptivity, which fall between online and offline RL. For example, in clinical trials, groups of patients undergo multiple treatments simultaneously and the treatment allocations are only be updated once the outcomes of the all the previous group have been observed (Yu et al., 2021). Similar settings with parallelized data-collection include marketing, advertising and numerical simulations (Esfandiari et al., 2021). There are also applications where the data collection needs to be validated prior to deployment by a human due to concerns of safety (Dann et al., 2019) and fairness (Koenecke et al., 2020), which with large scale data collection, is not feasible to have at every data point.

This motivates considering the multi-batch learning model (Perchet et al., 2015) where a data-set of size $n$ is collected in $K$ batches, and feedback from within a batch is only observed at the end of a batch. This means only data from previous batches can be used to select how the next batch is collected. This is also known as growing batch RL (Lange et al., 2012). We refer to it as *multi-batch RL* to avoid any confusion with offline RL, also called batch RL, which considers a single batch.

The multi-batch setting covers different levels of adaptivity to feedback, which refers to how frequently the learner can process feedback and use it to update its data collection strategy. We measure

it by the number of batches $K$. At one extreme, all the data is collected in a single batch ($K = 1$, offline-RL, no adaptivity). At the other extreme, each batch contains a single data-point ($K = n$, full adaptivity[1]). We say a learner is *adaptive* when $K > 1$. In some of the applications that parallelize data-collection mentioned above, algorithms with low-adaptivity where $K$ is as small as possible are of interest since there may be a cost associated to unifying and processing the parallelized data.

We study adaptivity in infinite-horizon discounted Markov Decision Processes (MDPs). We focus on 1) the policy evaluation (PE) problem, where the learner is tasked with estimating the value of a target policy, and 2) the best policy identification (BPI) problem, where the learner is tasked with finding a near-optimal policy. A desirable property of the learner is that it is **sample-efficient**, i.e. it only needs a dataset size that is polynomial in the dimension of the problem (e.g. state space size). We aim to understand the minimum level of adaptivity necessary for sample efficient learning.

**Linear Function Approximation:** MDPs faced in practice often have state spaces $\mathcal{S}$ or action spaces $\mathcal{A}$ that are infinite or too large to handle directly (Silver et al., 2016). Function approximation is used to reduce the learning problem to a smaller set of parameters that leverage structure in the problem. We consider a form of linear function approximation (Bellman et al., 1963) that assumes the (action)-value of a policy is a linear combination of known features of the state-action pair with an unknown parameter, each of dimension $d$. The sample complexity of algorithms is then measured with respect to the smaller dimension $d$ instead of the dimensions of the MDP ($|\mathcal{S}|, |\mathcal{A}|$).

**Adaptive vs Non-Adaptive:** It is known there is a sample-efficiency separation between offline RL ($K = 1$) and fully-adaptive RL ($K = n$) under linear function approximation. Algorithms have been shown to be sample-efficient in the fully-adaptive setting (Lattimore et al., 2020) and under various assumptions in the offline setting (Duan et al., 2020; Xie & Jiang, 2020). Without assumptions in the offline setting, it has been shown information-theoretically that no sample-efficient algorithms can solve the PE or BPI problem (Zanette, 2021) even under the best possible offline dataset, showing the separation. However, the MDP constructions of Zanette (2021) are easily solved with algorithms using $K = 2$ batches (see proofs of their Theorems 1 and 3), suggesting that the boundary of this separation may be between offline RL ($K = 1$) and RL with adaptivity ($K > 1$). This motivates studying the values of $K$ where sample-efficiency is not possible and asking the following questions:

*Does the boundary of sample-efficiency under linear function approximation lie between offline RL ($K = 1$) and RL with adaptivity ($K > 1$)? If not, is the boundary dimension-dependent?*

In this paper, we establish an $\Omega(\log \log d)$ lower-bound on the number of adaptive updates, $K$, required to solve both PE and BPI problems sample-efficiently. This answers the first question negatively and the second positively. This is achieved through a non-trivial extension of the framework of (Zanette, 2021) for the offline setting to the multi-batch setting, which we describe next.

**Learning Process:** When faced with an unknown MDP within a known class of MDPs, we consider a learner over $K$ rounds. In a round, the learner chooses a set of state-action queries with knowledge of the feedback from previous rounds. The following characteristics strengthen our lower bounds:

- Exact feedback: the feedback for a state-action query is the reward and transition *functions*, not a single sample. This makes a query equivalent to observing infinite samples from the reward and transition functions if these are stochastic and removes any hardness due to uncertainty.
- We consider two ways for the learner to specify the set of queries. The first (*policy-induced*) is through trajectories induced by chosen policies. The second (*policy-free*) explicitly specifies the state-action queries. Our results hold for any such set of queries whose size is polynomial in $d$.
- Realizability: the MDPs considered satisfy the linear representation of the action-values.

Tabular and finite-horizon MDPs and linear bandits are easily solved in the offline setting ($K = 1$) under this framework but infinite-horizon discounted MDPs are not (Zanette, 2021). Our work studies what happens beyond the offline setting ($K > 1$) for infinite-horizon discounted MDPs.

**Contributions:** Our results show that a number of batches $K$ constant with respect to the dimension $d$ is not enough to solve PE or BPI problems sample-efficiently. Specifically, we show that if we restrict the total number of queries (over all batches) to be polynomial in $d$ (sample-efficiency), then

---

[1]Typically, online refers to a setting with full-adaptivity along a single trajectory (sequence of transitions from a starting state, potentially with restarts). Our notion of full-adaptivity covers this, but is more general since we allow settings with a generative model where samples from any state-action pair can be drawn.

- there are PE problems that require $K = \Omega(\log\log d)$ to be solved to arbitrary accuracy.
- with only policy-free queries, there are PE and BPI problems that require $K = \Omega(\log\log d)$ to be solved to arbitrary accuracy, even if all policies satisfy linear realizability of their action-values.

These results show that adaptivity ($K > 1$) does not guarantee sample-efficiency. The level of adaptivity, or number of batches $K$, needed to guarantee sample-efficiency scales with the dimension $d$ of the linear representation. In particular, the boundary of sample-efficiency does not lie between offline RL ($K = 1$) and adaptive RL ($K > 1$). Instead this boundary must lie within a regime of adaptivity scaling with dimension: $\Omega(\log\log d) \leqslant K \leqslant n$.

Interestingly, the class of MDPs considered in Zanette (2021) can be solved with $d + 1$ queries and $K = 2$ batches (observing feedback from the first batch is enough to select queries in the second that fully solve the MDPs) [Zanette (2021), Theorem 4]. Our results show that this is not possible in general and that the class of MDPs we use for our results is fundamentally harder. From a technical perspective, our work uses tools from the theory of subspace packing with chordal distance (Soleymani & Mahdavifar, 2021). This enables the environment to erase information across multiple dimensions ($m$-dimensional subspaces, see Section 5) in response to queries, instead of a single direction as in Zanette (2021), which ultimately allows us to achieve lower-bounds for $K > 1$.

## 2 PRELIMINARIES

A Markov Decision Process (MDP) (Puterman, 1994) is a discrete-time stochastic process comprised of a set of states $\mathcal{S}$, a set of actions $\mathcal{A} = \bigcup_{s \in \mathcal{S}} \{\mathcal{A}_s\}$ where $\mathcal{A}_s$ is the action space in state $s \in \mathcal{S}$ and, for each state-action pair $(s, a) \in \mathcal{S} \times \mathcal{A}_s$, a next-state transition function given by a measure $p(\cdot|s, a)$ and a (deterministic) reward function $r(s, a) \in [-1, 1]$ (in fact, even the value functions defined below are in $[-1, 1]$ for our constructions). In a state $s$, an agent chooses an action $a$, receives a reward $r(s, a)$ and transitions to a new state according to $p(\cdot|s, a)$. Once in the new state, the process continues. The actions chosen by an agent are formalised by policies. A deterministic policy $\pi : \mathcal{S} \to \mathcal{A}$ is a mapping from a state to an action. In each state $s \in \mathcal{S}$, an agent following policy $\pi$ chooses action $\pi(s) \in \mathcal{A}_s$. We do not consider stochastic policies.

In this work, for a discount factor $\gamma \in [0, 1)$, we consider $\gamma$-discounted infinite-horizon MDPs. We measure the performance of a policy $\pi$ with respect to the value function $V^\pi : \mathcal{S} \to \mathbb{R}$,

$$V^\pi(s) = \mathbb{E}\Big[ \sum_{t=0}^{\infty} \gamma^t r(s_t, \pi(s_t)) | \pi, s_0 = s \Big],$$

where $s_t, a_t$ are the state and action in time-step $t$ and the expectation is with respect to the randomness in the transitions. This is a notion of long-term reward that describes the discounted rewards accumulated over future time-steps when following policy $\pi$ and starting in state $s$. We consider $\gamma$ as fixed throughout. It is also useful to work with the action-value function $Q^\pi : \mathcal{S} \times \mathcal{A} \to \mathbb{R}$,

$$Q^\pi(s, a) = \mathbb{E}\Big[ \sum_{t=0}^{\infty} \gamma^t r(s_t, \pi(s_t)) | \pi, s_0 = s, a_0 = a \Big],$$

which is similar to $V^\pi$, with the additional constraint of taking action $a$ in the first time-step. For a policy $\pi$, we define the Bellman evaluation operator $\mathcal{T}^\pi$ for action-value functions as:

$$(\mathcal{T}^\pi Q)(s, a) = r(s, a) + \gamma \mathbb{E}_{s' \sim p(\cdot|s,a)}[Q(s', \pi(s'))].$$

The action-value $Q^\pi$ of a policy $\pi$ is the unique fixed-point of the Bellman evaluation operator $\mathcal{T}^\pi$.

Under certain conditions on the state and action space, it is known that there exists a deterministic policy that simultaneously maximises $V^\pi$ and $Q^\pi$ for all states and actions [Puterman (1994), Theorem 6.2.12]. We call such a policy an optimal policy and denote it by $\pi^\star$. We will also denote $V^{\pi^\star} = V^\star$ and $Q^{\pi^\star} = Q^\star$. Given an MDP $M$, we will sometimes write $V_M^\pi$ to denote the value of a policy $\pi$ in the MDP $M$ (similarly for $V_M^\star, Q_M^\pi, Q_M^\star, p_M, r_M, \mathcal{T}_M^\pi$). We denote the unit Euclidean ball in $\mathbb{R}^d$ by $\mathcal{B} = \{x \in \mathbb{R}^d : \|x\|_2 \leqslant 1\}$ and its boundary by $\partial\mathcal{B} = \{x \in \mathbb{R}^d : \|x\|_2 = 1\}$. For $n$ vectors $v_1, ..., v_n \in \partial\mathcal{B}$, we denote the subspace spanning the vectors by $\langle v_1, ..., v_n \rangle$. For two positive functions $f$ and $g$, we say $f(x) = \Omega(g(x))$ if $\exists c > 0, N$ such that for all $x > N$, $f(x) \geqslant cg(x)$.

## 3 PROBLEM SETTING

In this section, we formally define the RL problems and the learning model we consider. We borrow the framework from the work of Zanette (2021) and extend it beyond the offline RL setting.

### 3.1 POLICY EVALUATION (PE)

Let $\mathcal{M}$ be a class of MDPs with the same $\mathcal{S}$ and $\mathcal{A}$. For $M \in \mathcal{M}$, let $\Pi_M$ be a set of (deterministic) policies and $\Pi = \{\Pi_M\}_{M \in \mathcal{M}}$. A PE problem defined by $(\mathcal{M}, \Pi)$ consists of:

1. **An instance** $(\bar{s}, M, \mathcal{M}, \pi_M, \Pi)$, where $M \in \mathcal{M}$ is an MDP, $\pi_M \in \Pi_M$ is a target policy and $\bar{s} \in \mathcal{S}$ is a starting state. $\mathcal{M}$, $\Pi$ and $\bar{s}$ are known but $M$ and $\pi_M$ are unknown.

2. **An interaction procedure** with the MDP $M$ to collect a dataset $\mathcal{D}$ (see Section 3.3).

3. **An objective:** Following the collection of the dataset $\mathcal{D}$, the target policy $\pi_M$ becomes known to the learner. Based on $\mathcal{D}$ and $\pi_M$, the learner produces an output $\widehat{Q}_{\mathcal{D}}(\bar{s}, \cdot)$ estimating the action-value $Q_M^{\pi_M}(\bar{s}, \cdot)$ of the target policy $\pi_M$. The performance of the learner is evaluated by the accuracy of the output on any instance, formalized as $(\varepsilon, \delta)$-soundness (Definition 3.1).

**Definition 3.1.** *A learner is $(\varepsilon, \delta)$-sound for PE problems characterised by $(\mathcal{M}, \Pi)$ if for all $M \in \mathcal{M}, \pi_M \in \Pi_M$, the learner faced with instance $(\bar{s}, M, \mathcal{M}, \pi_M, \Pi)$ outputs $\widehat{Q}_{\mathcal{D}}$ that is $\varepsilon$-accurate with probability[2] at least $1 - \delta$, i.e. it satisfies $\mathbb{P}\Big( \sup_{a \in \mathcal{A}} |Q_M^{\pi_M}(\bar{s}, a) - \widehat{Q}_{\mathcal{D}}(\bar{s}, a)| < \varepsilon \Big) > 1 - \delta$.*

### 3.2 BEST POLICY IDENTIFICATION (BPI)

Let $\mathcal{M}$ be a class of MDPs with the same $\mathcal{S}$ and $\mathcal{A}$. A BPI problem defined by $\mathcal{M}$ consists of:

1. **An instance** $(\bar{s}, M, \mathcal{M})$, where $M \in \mathcal{M}$ is an MDP in $\mathcal{M}$ and $\bar{s} \in \mathcal{S}$ is a starting state. $\mathcal{M}$ and $\bar{s}$ are known but $M$ is unknown.

2. **An interaction procedure** with the MDP $M$ to collect a dataset $\mathcal{D}$ (see Section 3.3).

3. **An objective:** Based on $\mathcal{D}$, the learner produces an output $\widehat{\pi}_{\mathcal{D}}$ of a near-optimal policy for $M$. The performance of the learner is evaluated by $(\varepsilon, \delta)$-soundness (Definition 3.2), i.e. the sub-optimality of the output policy on any instance (see 1.).

**Definition 3.2.** *A learner is $(\varepsilon, \delta)$-sound for BPI problems characterised by $\mathcal{M}$ if for all $M \in \mathcal{M}$, the learner faced with instance $(\bar{s}, M, \mathcal{M})$ outputs $\widehat{\pi}_{\mathcal{D}}$ that is $\varepsilon$-optimal with probability[2] at least $1 - \delta$, i.e. it satisfies $\mathbb{P}\Big( (V_M^{\star} - V_M^{\widehat{\pi}_{\mathcal{D}}})(\bar{s}) < \varepsilon \Big) > 1 - \delta$.*

### 3.3 MULTI-BATCH LEARNING MODEL

We define some important notions related to our learning model. A **query** is a state-action pair $(s, a) \in \mathcal{S} \times \mathcal{A}_s$ that is submitted to the unknown MDP $M$ and for which feedback is returned. A query formalises how the learner interacts with an MDP, the feedback received is defined next.

**Definition 3.3** (Query-Feedback). *In return for a query $(s, a)$ the environment provides feedback to the learner. For BPI, the feedback is the reward $r_M(s, a)$ and the transition function $p_M(\cdot|s, a)$. For PE, the learner also receives evaluations of the target policy $\pi_M$ for all states in the support of $p_M(\cdot|s, a)$, i.e. $\{\pi_M(s') : s' \in \mathcal{S} \text{ s.t. } p_M(s'|s, a) > 0\}$.*

**Remark 3.4.** *The learner receives the transition function $p_M(\cdot|s, a)$ for a query $(s, a)$, instead of a sample from $p_M(\cdot|s, a)$. This removes any statistical uncertainty and is equivalent to observing infinite samples, which strengthens any lower-bounds proven under this frame-work. The evaluations of the target policy $\pi_M$ are motivated by giving partial information about $\pi_M$ to the learner.*

We summarise the learning model in Algorithm 1. We consider two mechanisms for the learner to specify the set of queries $\mu_k$ at round $k$ (line 4). For both, we denote $n_k = |\mu_k|$ the number of queries at round $k$ and $n = \sum_{k=1}^{K} n_k$ the total number of queries.

---

[2]There is no randomness in the feedback of the dataset $\mathcal{D}$ (see Section 3.3) so the probabilities are with respect to randomness arising from the learner's query selection or output strategies.

---

**Algorithm 1** Multi-Batch Learning Model

---

1: (Input) PE or BPI problem and a number of batches $K$.
2: Initialise $\bar{\mathcal{D}}_0 = \varnothing$.
3: **for** $k = 1, ..., K$ **do**
4:    (Query Selection) With knowledge of $\bar{\mathcal{D}}_{k-1}$, learner chooses a set of queries $\mu_k$.
5:    (Data Collection) Environment receives queries $\mu_k$ and returns to learner $\mathcal{D}_k$ (set of queries + feedback for all queries in $\mu_k$ - see Definition 3.3). Learner updates $\bar{\mathcal{D}}_k = \bigcup_{i=1}^{k} \mathcal{D}_i$.
6: Set $\mathcal{D} = \bar{\mathcal{D}}_K \equiv \bigcup_{i=1}^{K} \mathcal{D}_i$ (and for PE, $\pi_M$ becomes known).
7: (Output) Learner returns $\widehat{Q}_{\mathcal{D}}$ (PE) or $\widehat{\pi}_{\mathcal{D}}$ (BPI).

---

**1. Policy-Free Queries:** In the first mechanism, the learner explicitly selects the set of queries $\mu_k$ by selecting a set of state-action pairs. The learner has access to the dataset $\bar{\mathcal{D}}_{k-1}$, which contains the feedback of the queries from the previous rounds from the MDP $M$ the learner is faced with. Let $\mathcal{M}_k \subset \mathcal{M}$ be the set of MDPs in $\mathcal{M}$ that would produce exactly the feedback in dataset $\bar{\mathcal{D}}_{k-1}$ given the queries in the previous rounds. Given the queries this is deterministic since there is no randomness in the feedback of a query (see Definition 3.3). The learner can use $\mathcal{M}_k$ in the selection of the queries at round $k$ but the specific MDP it is interacting with remains unknown if $|\mathcal{M}_k| > 1$.

**2. Policy-Induced Queries:** The second mechanism produces queries indirectly by selecting policies and using the queries along trajectories induced by these policies interacting with the MDP $M$. Stochastic transitions imply different realizations of a trajectory for a policy from a same starting-state. We allow the queries to be the state-actions pairs along all realizations of the trajectories.

**Definition 3.5** (Policy-Induced Queries (Zanette (2021), Definition 2))**.** *Fix a set $T_k = \{(s_{0i}^k, \pi_i^k, c_i^k)\}_{i=1}^{\kappa_k}$ of $\kappa_k$ triplets, each containing a starting state $s_{0i}^k$, a deterministic policy $\pi_i^k$ and a trajectory length $c_i^k$. Then the query set $\mu_k$ induced by $T_k$ is defined as*

$$\mu_k = \bigcup_{(s_0, \pi, c) \in T_k} Reach(s_0, \pi, c)$$

*where* $\quad Reach(s_0, \pi, c) = \{(s, a) | \exists t < c \text{ s.t. } \mathbb{P}((s_t, a_t) = (s, a) | \pi, s_0) > 0\}$

*are the state-action pairs reachable in $c$ or less time-steps from $s_0$ using policy $\pi$. Note $(s_t, a_t)$ is the random state-action pair encountered at time-step $t$ upon following $\pi$ from $s_0$.*

The learner specifies a set $T_k$ from which a set of queries $\mu_k$ is induced[3]. Similarly to policy-free queries, the learner can use $\mathcal{M}_k$ in the selection of $T_k$ at round $k$ but the specific MDP it is interacting with remains unknown if $|\mathcal{M}_k| > 1$.

**Remark 3.6.** *Policy-induced queries include policy-free queries as a special case ($c_i^k = 1$). However, if the dynamics of all MDPs in the class $\mathcal{M}$ are the same, then policy-induced queries are also policy free because the learner knows the dynamics of the MDP it is interacting with. So it knows exactly the queries that any set $T_k$ will induce and can specify these as policy-free queries. If the dynamics of all MDPs in the class $\mathcal{M}$ are not the same, then policy-induced queries can reveal more information because a trajectory is guided by the dynamics of the MDP while policy-free queries only reveal information about individual distinct transitions. Because of this, we will obtain slightly stronger results for policy-free queries in Section 4.*

**Adaptivity:** The multi-batch learning model encompasses different levels of adaptivity to feedback, which is measured by the number of batches $K$:

• For $K = 1$, the learner is **non-adaptive** and the dataset $\mathcal{D}$ is collected in a single batch. This is the model considered by Zanette (2021) for offline RL that we extend for general $K$.

• For $K > 1$, the learner is **adaptive** and the dataset $\mathcal{D}$ is collected in multiple batches. Queries for a batch are selected based on feedback from previous batches.

• For $K = n$ (i.e. each batch contains a single data-point), the learner is **fully-adaptive**. The queries are selected sequentially and depend on the feedback from previous queries.

---

[3]A sample-efficient learner requires $|\mu_k|$ to be polynomial in $d$ but the learner does not directly select $\mu_k$. For example, if the transition function is stochastic it is possible that $T_k$ induces a $\mu_k$ with $|\mu_k| = \infty$ or non-polynomial in dimension. The MDPs we consider all have deterministic transitions so this is not an issue.

## 4 MAIN RESULTS

In this section, we present our main results: lower-bounds on the number of rounds $K$ for sample-efficient algorithms. First, we state some assumptions. We assume $\gamma > \sqrt{3/4}$. We also consider a linear representation of the action-values for a known feature map of state-action pairs. This is a form of linear function approximation that is strictly more general than linear MDPs (Zanette et al., 2020), which assume the reward and transition functions are linearly representable. Specifically:

**Assumption 4.1** ($Q_M^\pi$-Realizability (Zanette (2021), Assumption 1)). *Given any PE problem instance* $(\bar{s}, M, \mathcal{M}, \pi_M, \Pi)$*, there exists a known feature map* $\phi : \mathcal{S} \times \mathcal{A} \to \mathbb{R}^d$ *s.t.* $\|\phi(\cdot, \cdot)\|_2 \leqslant 1$ *and there exists* $\theta_M^{\pi_M} \in \mathcal{B}$ *such that for all* $(s, a) \in \mathcal{S} \times \mathcal{A}$,

$$Q_M^{\pi_M}(s, a) = \phi(s, a)^T \theta_M^{\pi_M}.$$

**Assumption 4.2** ($Q^\pi$-Realizability for every policy (Zanette (2021), Assumption 3)). *Given any BPI* $(\bar{s}, M, \mathcal{M})$ *or PE* $(\bar{s}, M, \mathcal{M}, \pi_M, \Pi)$ *problem instance, there exists a known feature map* $\phi : \mathcal{S} \times \mathcal{A} \to \mathbb{R}^d$ *s.t.* $\|\phi(\cdot, \cdot)\|_2 \leqslant 1$ *and for any policy* $\pi$ *there exists* $\theta_M^\pi \in \mathcal{B}$ *s.t. for all* $(s, a) \in \mathcal{S} \times \mathcal{A}$,

$$Q_M^\pi(s, a) = \phi(s, a)^T \theta_M^\pi.$$

This first assumption is used for PE problems. The second is stronger as it concerns the action-value of every policy (not just policies in $\Pi$) and in particular it holds for $Q^\star$. We assume the learner is aware when these assumptions hold. Finally, we formally define a sample-efficient learner:

**Definition 4.3.** *A learner for PE or BPI problems under Assumptions 4.1 or 4.2 is sample-efficient if its total number of queries* $n = \sum_{k=1}^K n_k$ *is polynomial in* $d$.

### 4.1 POLICY-INDUCED QUERIES

We first present a result for PE under policy-induced queries. Since all MDPs in the class used in the proof share the same dynamics, policy-induced queries are equivalent to policy-free queries (see Remark 3.6) and the result holds for both. The full proof can be found in Appendix E.

**Theorem 4.4.** *Fix* $d$ *sufficiently large. There exists a class of MDPs* $\mathcal{M}$ *and policies* $\Pi$ *defining PE problems* $(\bar{s}, M, \mathcal{M}, \pi_M, \Pi)$ *satisfying Assumption 4.1 such that any sample-efficient learner better than* $(1, 1/2)$*-sound using policy-induced or policy-free queries requires* $K = \Omega(\log \log d)$.

In the class of MDPs used for Theorem 4.4 we can hide information about $M \in \mathcal{M}$ in the target-policy for PE but cannot for BPI. Instead, we could hide information in the transitions but this can be revealed by following policy trajectories (policy-induced queries) in our constructions. In the next section, we restrict the learner to policy-free queries and provide results for both PE and BPI. Note that a $(1, 1/2)$-sound learner performs poorly since the MDP class has value functions in $[-1, 1]$.

### 4.2 POLICY-FREE QUERIES

We now consider only policy-free queries, which gives the environment freedom to hide information in the transition function of the MDP and leads to lower-bounds for PE and BPI that hold for the stronger Assumption 4.2 of all-policy realizability. The full proof can be found in Appendix F.

**Theorem 4.5.** *Fix* $d$ *sufficiently large. There exists a class of MDPs* $\mathcal{M}$ *and policies* $\Pi$ *defining problems for PE* $(\bar{s}, M, \mathcal{M}, \pi_M, \Pi)$ *and BPI* $(\bar{s}, M, \mathcal{M})$ *satisfying Assumption 4.2 such that any sample-efficient learner better than* $(1, 1/2)$*-sound using policy-free queries requires* $K = \Omega(\log \log d)$.

### 4.3 DISCUSSION

The results indicate that $K = \Omega(\log \log d)$ batches are necessary for solving PE or BPI tasks sample-efficiently under realizable linear function approximation. Beyond the exact dependence on $d$, the significance of these results is that just having $K > 1$ is insufficient. In particular, more adaptivity is needed as the dimension of the linear representation increases. These results demonstrate that sample efficiency is impossible not only in offline RL, but also in settings with some level of adaptivity ($K = o(\log \log d)$). Therefore, the boundary at which sample-efficiency becomes impossible is at a

dimension-dependent level of adaptivity between offline and full-adaptivity. This leaves interesting open directions on the existence of a sample-efficient algorithm using $K = O(\log \log d)$ batches.

Our lower-bounds no longer hold if the action space $\mathcal{A}$ is finite or with coverage assumptions on the collected data (for BPI). While it may be necessary to have coverage assumptions in an offline setting, this assumption need not be made when data is collected adaptively, as in our setting.

Furthermore, in Appendix B we provide results for the fully adaptive setting where the number of batches $K$ is equal to the number of total queries $n$ (i.e. each batch contains a single query). We show that if the feature-space covers $\mathcal{B}$, there is a learner that solves any realizable (Assumption 4.1) PE problem in $d$ queries. Assuming a known target policy $\pi$, a linear dependence on $d$ was already known to be possible using roll-outs from $\pi$ (Lattimore et al., 2020), however the dependence on $d$ was coupled with other quantities such as the effective horizon $1/(1-\gamma)$ and desired accuracy $\varepsilon$. Our result states that $d$-queries are sufficient to find $Q^\pi$ exactly ($\varepsilon = 0$), independently of $\gamma$. Our result relies heavily on the condition that the learner observes the transition function $p_M(\cdot|s,a)$ rather than a sample $s' \sim p_M(\cdot|s,a)$ for a query $(s,a)$ (see Section 3.3), though our result under this condition is strong since using this condition with the roll-outs from $\pi$ would not give exact convergence in $d$-queries. We also provide a matching lower-bound (that also holds for BPI). These results serve to illustrate the trade-off between sample-efficiency and low-adaptivity for PE under our framework. Full-adaptivity allows low sample-complexity, while reducing adaptivity below $K = o(\log \log d)$ comes at the cost of high sample-complexity (losing sample-efficiency).

## 4.4 RELATED WORKS

The works discussed below are for infinite-horizon discounted MDPs unless stated otherwise.

**Tabular MDPs** ($|\mathcal{A}|, |\mathcal{S}|$ are "small") can be solved in the offline setting under policy-free queries: model-based approaches (Li et al., 2020; Agarwal et al., 2020) under a generative model are minimax-optimal (Azar et al., 2013) with sample-complexity linear in the dimension of the MDP $|\mathcal{A}| \times |\mathcal{S}|$. These methods estimate the MDP by sampling equally from all state-action pair and then use dynamic programming approaches on the estimated MDP. In particular, all the samples are drawn in a single-batch. Beyond the generative model and under a restricted form of our policy-induced queries (Definition 3.5), tabular offline RL is no longer sample efficient (Xiao et al., 2022) and requires a number of samples exponential in $|\mathcal{S}|$ or $1/(1 - \gamma)$.

**Linear Function Approximation:** In the offline setting, there are lower-bounds showing OPE or BPI is not possible with samples polynomial in the effective horizon $1/(1 - \gamma)$ or the linear dimension (Amortila et al., 2020; Chen et al., 2021; Zanette, 2021). The bound of Zanette (2021) is the strongest as it holds for any data-distribution. These exponential lower-bounds can be overcome with assumptions such as low-distribution shift (Chen et al., 2021), low inherent Bellman-error (Xie & Jiang, 2020; Duan et al., 2020) or low local inherent Bellman error (Zanette, 2023). We refer the reader to the work of Zanette (2021) for a more in depth discussion of offline RL. In the fully-adaptive setting, there are sample-efficient algorithms under all-policy realizability (Lattimore et al., 2020), under $V^\star$-realizability only (Weisz et al., 2021) (if the action space is finite) and under linear MDPs (Taupin et al., 2023; Kitamura et al., 2023).

**The multi-batch learning model** has been studied extensively for bandit algorithms (Perchet et al., 2015; Jun et al., 2016; Gao et al., 2019; Esfandiari et al., 2021; Duchi et al., 2018; Han et al., 2020; Ruan et al., 2021). In RL, it has been studied in the regret-minimisation setting for finite-horizon tabular (Zihan et al., 2022) and linear MDPs (Wang et al., 2021) and MDPs under general function approximation (Xiong et al., 2023). A closely related notion is deployment efficiency (Matsushima et al., 2021), which constrains batches to be of a fixed size consisting of trajectories from a single policy. In finite-horizon linear MDPs, it has been shown that BPI can be solved to arbitrary accuracy with a number of deployments independent of the dimension $d$ (Huang et al., 2022; Qiao & Wang, 2023) where the deployed policy is a finite mixture of deterministic policies. Our results suggest that infinite-horizon discounted MDPs under more general linear representation of action-values are fundamentally harder since the number of deployments must scale with dimension.

Please refer to Appendix A for a discussion of works related to low-switching cost (Qiao et al., 2022; Bai et al., 2019; Zhang et al., 2020; Gao et al., 2021; Wang et al., 2021; Qiao & Wang, 2023; Qiao et al., 2023) and policy fine-tuning (Xie et al., 2021; Zhang & Zanette, 2023).

## 5 PROOF SKETCH

In this section, we provide intuition for the proof of Theorem 4.4. We extend the ideas of Zanette (2021) beyond offline RL to our multi-batch problem (see the end of this section for a comparison). We consider the PE problem with policy-free queries under Assumption 4.1 for feature vectors $\phi(\cdot, \cdot)$ covering the unit Euclidean ball $\mathcal{B}$ (see Figure 1). The intuition for Theorem 4.5 is closely related.

Consider the first batch of data with $n_1$ queries and let $(s_i, a_i)$ be the i-th query and $(s_i^+, \pi_M(s_i^+))$ the corresponding (assumed deterministic) successor state and target policy evaluation. Define

$$\Phi = \begin{bmatrix} \phi(s_1, a_1)^T \\ ... \\ \phi(s_{n_1}, a_{n_1})^T \end{bmatrix}, \quad r = \begin{bmatrix} r(s_1, a_1) \\ ... \\ r(s_{n_1}, a_{n_1}) \end{bmatrix}, \quad \Phi^+ = \begin{bmatrix} \phi(s_1^+, \pi_M(s_1^+)^T \\ ... \\ \phi(s_{n_1}^+, \pi_M(s_{n_1}^+))^T \end{bmatrix}.$$

Since $Q^{\pi_M}$ is the fixed point of the Bellman evaluation operator: $Q^{\pi_M}(s, a) = (\mathcal{T}^{\pi_M} Q^{\pi_M})(s, a)$ and by Assumption 4.1 there exists $\theta_M^{\pi_M}$ such that $Q^{\pi_M}(s, a) = \phi(s, a)^T \theta_M^{\pi_M}$ for any $(s, a)$, the learner aims to find a solution $\theta$ satisfying the (local) Bellman equation

$$\Phi\theta = r + \gamma\Phi^+\theta \implies (\Phi - \gamma\Phi^+)\theta = r.$$

If $X = \Phi - \gamma\Phi^+$ is not full-rank, this equation does not have a unique solution. The learner only chooses $\Phi$. The environment, with knowledge of $\Phi$, can pick $\Phi^+$ to maximise the dimension of the null-space[4] of $X$, which can be viewed as erasing information along many directions (see Figure 1 left). This phenomenon where the value of a policy in a state depends on the same value in the successor states is known as bootstrapping and is the mechanism inducing hardness in our setting as it allows the environment to choose these successor states adversarially to erase information.

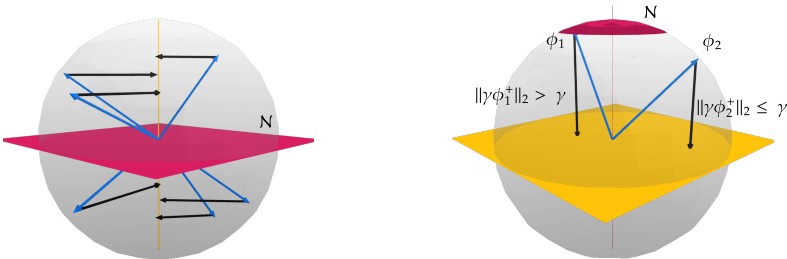

Figure 1: Left: **Information can be erased in multiple directions:** Consider the setting where information is being erased along the pink plane $\mathcal{N}$: the learner's queries $\phi(s_i, a_i)$ are shown in blue and the environment's responses $-\gamma\phi(s_i^+, \pi_M(s_i^+))$ are shown in black. The rows of $X$, $\phi(s_i, a_i) - \gamma\phi(s_i^+, \pi_M(s_i^+))$ (blue + black vectors) all lie on the yellow line so the learner acquires no information in the directions of the pink subspace $\mathcal{N}$, the null-space of $X$. Right: **Information cannot be erased in all directions:** Consider the opposite setting where information is being erased along the pink line $\mathcal{N}$. Because of the constraint $\|\phi(s, a)\|_2 \leq 1$ and $\gamma < 1$, a query $\phi_1$ (in blue) in the pink cap cannot have $\phi_1 - \gamma\phi_1^+$ (blue + black vector) projected back onto the yellow plane (unless $\|\gamma\phi_1\|_2 > \gamma \implies \|\phi_1\|_2 > 1$). Despite the environment not being able to erase information in certain directions, if the number of queries is "small", it can always find directions to erase.

Although the learner can prevent the environment from erasing information along certain directions (see Figure 1 right), since $n_1$ is polynomial in $d$ this only prevents the environment from erasing information in a limited number of directions. Specifically, we show that if $n_1$ is less than exponential in $d^{1/4}$ then there is a subspace of dimension $d^{1/4}$ that can be included in the null-space of $X$.

Prior to choosing $n_2$ queries for the 2nd batch, the learner observes the feedback from the first batch. It becomes aware of the directions of the null-space and can focus its queries for the next round on these directions. However, the null-space is still at least $d^{1/4}$-dimensional and so the same

---

[4]By the rank-nullity theorem, this is equivalent to minimising the rank of $X$.

reasoning as in the first round can be applied where the original dimension is now $d^{1/4}$. So if $n_2$ is less than exponential in $d^{1/16}$ then there is a subspace of dimension $d^{1/16}$ that can be included in the null-space of the new local Bellman equation that includes the data from the 1st and 2nd batch.

After $k$ rounds if the number of queries at round $k$, $n_k$ is less than exponential in $d^{1/4^k}$, then the null-space of the local Bellman equation is still at least $d^{1/4^k}$-dimensional. If $\exp(d^{1/4^k})$ is more than polynomial in $d$, the sample-efficient learner cannot prevent a non-zero null-space and the problem cannot be solved. The learner must reach a round $K$ where $\exp(d^{1/4^K})$ becomes polynomial in $d$, requiring $K = \Omega(\log \log d)$ rounds, from which we get our lower-bound.

**Description of the MDP construction** for which the learner cannot do better than $(1, 1/2)$-soundness. Consider an MDP class $\mathcal{M}$ with $\mathcal{S} = \mathcal{A} = \mathcal{B}$ and a feature map $\phi$ such that $\phi(s, a) = a$. The successor state of $(s, a)$ is deterministic and is the action $a$. Fix $w \in \partial \mathcal{B}$ and consider two MDPs: $M_{w,+}$ and $M_{w,-}$. We denote the reward function for either MDP with the same subscript:

$$\text{for } z \in \{+, -\}: \quad \text{on } M_{w,z}: \quad r_{w,z}(s, a) = \begin{cases} 0, & \text{if } a \notin \mathcal{C}_\gamma(w) \cup \mathcal{C}_\gamma(-w) \\ z(1 - \gamma)a^T w, & \text{otherwise,} \end{cases}$$

where $\mathcal{C}_\gamma(w) = \{x \in \mathcal{B} : x^T w / \|w\|_2 > \gamma\}$ is the $\gamma$-hyperspherical cap of $w$. For a $w \in \partial \mathcal{B}$ and a carefully designed target policy $\pi_w$, we show that Assumption 4.1 holds. We also show with the reasoning described above that if $n_k = poly(d)$ for all $k$ and $K = o(\log \log d)$, then none of the learner's queries are in $\mathcal{C}_\gamma(w) \cup \mathcal{C}_\gamma(-w)$. Therefore the feedback observed contains no information about the sign of the rewards in $\mathcal{C}_\gamma(w) \cup \mathcal{C}_\gamma(-w)$. $M_{w,+}$ is indistinguishable from $M_{w,-}$. Since $Q^{\pi_w}_{M_{w,+}}(\bar{s}, w) = 1$ and $Q^{\pi_w}_{M_{w,-}}(\bar{s}, w) = -1$, the learner must incur an error of 1 with probability at least $1/2$. All the details are in Appendix E, including further illustrations in Appendix E.5. The construction for Theorem 4.5 is similar but the transitions differ across MDPs (see Appendix F).

**To improve the lower-bound with the current construction**, we require the existence of a subspace packing (of size exponential in $d$ and subspace dimension $cd$ (instead of $d^{1/4}$), for some $c < 1$) with minimal chordal distance (see Appendix D) between subspaces $d_{\min}(\mathcal{C}) \geqslant \sqrt{d - (2\gamma^2 - 1)^2}$.

After $k$ rounds, information would be missing along a subspace of dimension $c^k d$, (instead of $d^{1/4^k}$) from which we could get a $\log d$ lower-bound. As far as we are aware, such a result is not available in the literature, nor is a subspace covering result that would rule out the possibility of such a packing. We highlight that our MDP construction can be combined with any subspace packing result, paving the way for improved lower-bounds should new subspace packing procedures be developed.

**Comparison to Zanette (2021):** The work of Zanette (2021) erases information along a 1-dimensional subspace ($X$ is of rank $d - 1$). Therefore after having observed one round of feedback, a single additional query is sufficient to distinguish the MDP and solve the problem. Our constructions erase information along $m$-dimensional subspaces where we attempt to maximise $m$ so that even after having observed feedback revealing this subspace, a polynomial number of queries remains insufficient to distinguish the MDP, more adaptive rounds are needed.

## 6 CONCLUSION

In this work, we have studied the connection between adaptivity and sample-efficiency for RL algorithms solving PE and BPI problems under $d$-dimensional linear function approximation. For multi-batch learning, we have established $\Omega(\log \log d)$ lower-bounds on the number of batches $K$ needed to solve the RL problems sample-efficiently (number of queries polynomial in $d$). In particular, having adaptivity ($K > 1$) does not guarantee sample-efficiency. Consequently the boundary of sample efficiency must not lie between batch RL ($K = 1$) and adaptive RL ($K > 1$) but rather within a regime of adaptivity scaling with dimension. These insights contribute to a deeper understanding of the trade-offs and possibilities in designing sample-efficient RL algorithms with low-adaptivity.

It remains unclear if the $\log \log d$ dependence on $d$ is tight. An upper-bound similar to the one we have given for the fully-adaptive PE problem in Appendix B could be established by developing new tools in the theory of subspace covering. These would formalise the number of directions for which the learner can prevent information erasure. It is also unclear if Theorem 4.4 under policy-induced queries also holds for BPI or if the sample-efficiency of BPI-algorithms in low-adaptivity settings differs for policy-induced and policy-free queries. We leave these as future work.

ACKNOWLEDGMENTS AND DISCLOSURE OF FUNDING

EJ is funded by EPSRC through the Modern Statistics and Statistical Machine Learning (StatML) CDT (grant no. EP/S023151/1) and thanks G-Research for partly funding attendance to ICLR 2024.

We would like to thank the reviewers and meta-reviewers for their time and feedback, which led to a better version of this paper. We would also like to thank Nathan Mankovich and Hugo Chu for helpful discussions.

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

## A   FURTHER RELATED WORKS

The works discussed below are for infinite-horizon discounted MDPs unless stated otherwise.

**Low-Switching Cost:** Limited adaptivity in RL has mostly been studied in the context of regret-minimisation algorithms with low-switching cost for finite-horizon episodic MDPs, i.e. minimising the number of times the policy used changes from one episode to the next. These works are not directly comparable because they study regret-minimisation for finite-horizon MDPs and we study BPI and PE in the discounted setting. Nevertheless, there are works on tabular MDPs (Qiao et al., 2022; Bai et al., 2019; Zhang et al., 2020), linear MDPs (Gao et al., 2021; Wang et al., 2021; Qiao & Wang, 2023) and MDPs with a linear representation for the action values (Qiao et al., 2023).

**The policy finetuning** setting assumes access to an offline dataset that can be complemented with online trajectories (Xie et al., 2021) but is different from our setting since there is no adaptivity constraint in the online algorithm, i.e. once the initial dataset has been collected, the query selection strategy can be updated after each new observation (or episode in the episodic setting). However, if the additional trajectories are collected using a non-adaptive policy instead of an online algorithm, we can recover our setting with $K = 2$ batches. This is studied by Zhang & Zanette (2023) who show that for finite-horizon tabular MDPs, $K = 2$ is enough to solve the BPI problem to arbitrary accuracy. Our results rule out achieving a similar result for infinite-horizon discounted MDPs under policy-free queries and linear function approximation.

## B   BOUNDS FOR THE FULLY ADAPTIVE SETTING

In this section, we show an upper-bound result for the fully adaptive setting ($K = n$ - see Section 3.3). In this setting, the oracle selects one query in each round or batch of data, so chooses $(s_k, a_k)$ at round $k$ with knowledge of the feedback from queries up to time $k - 1$. The number of rounds $K$ coincides with the number of queries $n$ (each batch contains one query). In particular, since it is one query at a time, there is no difference between policy-induced or policy-free queries. The upper bound we show relies on the following assumptions on the feature space:

**Assumption B.1** (Feature Map). *Fix a feature map $\phi$. Given any orthonormal set of vectors $\{u_1, ..., u_n\}$ with $n < d$, it is possible to choose a state-action pair $(s, a)$ such that $\phi(s, a) \in \langle u_1, ..., u_n \rangle^\perp$ and $\|\phi(s, a)\|_2 = 1$.*

The superscript $\perp$ on a subspace refers to the orthogonal complememnt of the subspace.

**Theorem B.2.** *Fix $d > 0$. Consider any PE problem $(\bar{s}, M, \mathcal{M}, \pi_M, \Pi)$ satisfying Assumptions 4.1 and B.1 for the same feature map $\phi$, then there exists a fully-adaptive learner that solves the PE problem exactly in at most $d$ queries.*

We complement our upper-bound with a matching lower-bound, which holds for both PE and BPI.

**Theorem B.3.** *Fix $d > 0$. There exists a class of MDPs $\mathcal{M}$ and target policies $\Pi$ characterising PE problems $(\bar{s}, M, \mathcal{M}, \pi_M, \Pi)$ and BPI problems $(\bar{s}, M, \mathcal{M})$ that satisfy Assumption 4.2 and share the same $\bar{s}$ and $M$ such that any fully-adaptive learner that is better than $(1, 1/2)$-sound requires $K = n \geqslant d$.*

Theorem B.2 together with Theorem B.3 shows that exactly $d$ queries are optimal for solving a PE problem under our feedback model and Assumption B.1. The lower-bound is to be expected because the learner is operating in a $d$-dimensional feature space so has to learn in $d$ directions. However, it is interesting that the structure imposed by Assumption B.1 on the learner's capacity to explore the feature space is sufficient for the learner to fully solve the problem in only $d$ queries. A similar assumption was studied by Jia et al. (2023) to obtain a sample-efficient algorithm for BPI in finite-horizon MDPs. We can obtain this result for PE with a simple analysis because the learner can exploit that the action-value of a policy is the fixed point of a linear operator, the Bellman evaluation operator. An equivalent approach does not work for BPI since the action-value of the optimal policy is not the fixed point of a linear operator.

The proofs of the theorems in this section can be found in Appendix G.

## C  HYPER-SPHERICAL CAPS AND SECTORS FOR SUBSPACES

Recall that $\mathcal{B} = \{x \in \mathbb{R}^d : \|x\|_2 \leqslant 1\}$ is the $d$-dimensional unit hyper-sphere.

**Definition C.1.** *Fix $w \in \mathcal{B}$. Define the $\gamma$-hyperspherical cap of $w$ as:*

$$\mathcal{C}_\gamma(w) = \left\{ x \in \mathcal{B} : \frac{x^T w}{\|w\|_2} > \gamma \right\}.$$

A vector $x$ is in the $\gamma$-hyperspherical cap of $w$ if the angle $\theta$ between $x$ and $w$ satisfies

$$\gamma < \|x\|_2 \cos\theta \iff \theta < \arccos\left(\frac{\gamma}{\|x\|_2}\right).$$

With $\gamma$ close to $1$, these represent a set of vectors in the hyper-cone around $w$ that are close to the boundary of $\mathcal{B}$ (see Figure 1 right). The key property that motivates considering vectors in this set is that they require a vector of norm greater than $\gamma$ to be projected in a direction orthogonal to $w$. We extend the notion of $\gamma$-hyperspherical caps to subspaces of multiple dimensions.

**Definition C.2.** *Fix a subspace $H$ of $\mathbb{R}^d$. Define the $\gamma$-hyperspherical sector of $H$ as*

$$\overset{\triangle}{\mathcal{C}}_\gamma(H) = \left\{ x \in \mathcal{B} : \exists v \in H \text{ s.t. } \frac{x^T v}{\|v\|_2 \|x\|_2} > \gamma \right\}.$$

*See Figure 2 for an illustration.*

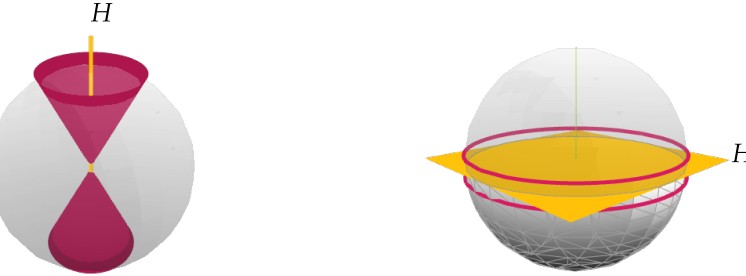

Figure 2: Illustration of a hyperspherical sector of a 1-dimensional subspace (left) and a 2-dimensional subspace (right - all vectors whose direction is within the two pink bands). In both cases, the subspace $H$ is in yellow.

Beyond the extension to subspaces of multiple dimensions, Definition C.2 differs from Definition C.1 in two ways. It is a hyper-spherical sector rather than cap, which does not restrict the vectors to be close to the boundary of $\mathcal{B}$. It is two sided meaning that if $x \in \overset{\triangle}{\mathcal{C}}_\gamma(H)$, then $-v \in \overset{\triangle}{\mathcal{C}}_\gamma(H)$. Note that the subspace $H$ is defined on $\mathbb{R}^d$ but $\overset{\triangle}{\mathcal{C}}_\gamma(H) \subset \mathcal{B}$.

Similar to the intuition in the 1-dimensional case, a vector $x$ is in $\overset{\triangle}{\mathcal{C}}_\gamma(H)$ if there is a vector in $H$ whose angle with $x$ is "small". It is equivalent to taking the unions of the 1-dimensional $\gamma$-hyperspherical sectors of all the vectors in $H$. Again, the key property that motivates considering vectors in this set is that they require a vector of norm greater than $\gamma$ to be projected in a direction orthogonal to $H$.

We also note that a vector $x$ is not in $\overset{\triangle}{\mathcal{C}}_\gamma(H)$ if the subspace $H$ does not intersect $\overset{\triangle}{\mathcal{C}}_\gamma(\langle x \rangle)$, the $\gamma$-hyperspherical sector of $\langle x \rangle$. This allows us to work with the $\gamma$-hyperspherical sectors of one dimensional subspaces (span of a single vector) instead of subspaces of multiple dimensions. We will also abuse notation and write $\overset{\triangle}{\mathcal{C}}_\gamma(x)$ instead of $\overset{\triangle}{\mathcal{C}}_\gamma(\langle x \rangle)$

# D  SUBSPACE PACKING

## D.1  PRELIMINARIES

Denote the set of all $m$-dimensional subspaces of $\mathbb{R}^d$ as $\mathcal{G}_{m,d}(\mathbb{R})$, which is called the **Grassmannian space**. An element $A \in \mathcal{G}_{m,d}(\mathbb{R})$ is an $m$-dimensional subspace of $\mathbb{R}^d$.

We present a measure of distance between subspaces known as the **chordal distance** (Conway et al., 1996). Fix two subspaces $A, B \in \mathcal{G}_{m,d}(\mathbb{R})$. The principal angles $\theta_1, ..., \theta_m \in [0, \pi/2]$ between $A$ and $B$ are defined as

$$\cos \theta_i = \max_{a \in A} \max_{B \in B} \frac{a^T b}{\|a\|_2 \|b\|_2} = a_i^T b_i,$$

for $i = 1, ..., m$ such that $\|a_i\|_2 = \|b_i\|_2 = 1$, $a^T a_j = 0$, $b^T b_j = 0$ for $1 \leqslant j \leqslant i - 1$. The chordal distance $d_c$ is then defined as

$$d_c(A, B) = \sqrt{\sin^2 \theta_1 + \sin^2 \theta_2 + ... + \sin^2 \theta_m}.$$

We define the notion of subspace packing, which is the usual notion of a packing where the set is the Grassmanian space $\mathcal{G}_{m,d}(\mathbb{R})$ and the distance is the chordal distance.

**Definition D.1.** *A subspace packing $\mathcal{C}$ of $\mathcal{G}_{m,d}(\mathbb{R})$ is a set of $m$-dimensional subspaces in $\mathbb{R}^d$ of size $|\mathcal{C}|$, i.e. it is a subset of $\mathcal{G}_{m,d}(\mathbb{R})$. The minimum distance between elements of $\mathcal{C}$ is measured by the chordal distance and is denoted*

$$d_{\min}(\mathcal{C}) = \min_{A, B \in \mathcal{C}, A \neq B} d_c(A, B).$$

## D.2  A SUBSPACE PACKING BOUND

**Lemma D.2** (Soleymani & Mahdavifar (2021), Theorem 4). *Fix $d = 2^N$ for some $N \in \mathbb{N}$ and integers $k < d$ and $m < d$. There exists a packing $\mathcal{C}$ in $G_{m,d}(\mathbb{R})$ of size $|\mathcal{C}| = \left(\frac{d}{2}\right)^{\lceil k/2 \rceil - 1} \lfloor \frac{d}{m} \rfloor$ s.t*

$$d_{\min}(\mathcal{C}) \geqslant \sqrt{m} \sqrt{1 - \frac{m(k-1)^2}{d}}.$$

Theorem 4 from Soleymani & Mahdavifar (2021) is presented for packings in $G_{m,d}(\mathbb{C})$ but they give (in Remark 1) a mapping from a packing in $G_{m/2,d/2}(\mathbb{C})$ to a packing in $G_{m,d}(\mathbb{R})$ that preserves the normalized distance $\delta_c$ ($= d_{\min}(\mathcal{C})/\sqrt{m}$) between the elements of the packing, giving the result presented above.

## D.3  EXISTENCE OF AN ISOLATED SUBSPACE

The following lemma is the key result for the construction of the class of MDPs used in the proof of our main results. It establishes that if a number of points $n$ is "small", then for any $n$ points there exists a subspace whose $\gamma$-sector contains none of the $n$ points. The environment can erase information along this subspace (see Section 5). See Appendix C for the definition of $\overset{\triangle}{\mathcal{C}}_\gamma(H)$ for a subspace $H$. The proof is given in Appendix D.3.2.

Throughout, we will use $g(\gamma) = 2\gamma^2 - 1$.

**Lemma D.3.** *Fix $d = 2^N$ for some $N \geqslant 8$. Consider $D = \{y_1, ..., y_n\}$, a set of $n$ points s.t. $y_i \in \mathcal{B}$ for all $i \in [n]$. If*

$$n + 1 \leqslant \left(\frac{d}{2}\right)^{\frac{1}{8} g(\gamma) d^{1/4}},$$

*then there exists a subspace $A \in \mathcal{G}_{2^{\lceil N/4 \rceil}, d}(\mathbb{R})$ of dimension $2^{\lceil N/4 \rceil}$ s.t.*

$$\forall x \in D, \qquad x \notin \overset{\triangle}{\mathcal{C}}_\gamma(A).$$

### D.3.1 Preliminary Lemmas

The proof of Lemma D.3 relies on the following lemmas. See Appendix D for the definition of $\mathcal{G}_{m,d}(\mathbb{R})$ and the definition of a subspace packing $\mathcal{C}$.

**Lemma D.4.** *Fix $d = 2^N$ for some $N \in \mathbb{N}$. If there exists a packing $\mathcal{C}$ in $G_{m,d}(\mathbb{R})$ of size $|\mathcal{C}| \geqslant n+1$ s.t. $d_{\min}(\mathcal{C}) \geqslant \sqrt{m - g(\gamma)^2}$, then given a set $\{y_1, ..., y_n\}$ of $n$ queries s.t. $y_i \in \mathcal{B}$ for all $i \in [n]$, there exists a subspace $H \in G_{m,d}(\mathbb{R})$ of dimension $m$ s.t. $y_i \notin \overset{\triangle}{\mathcal{C}}_\gamma(A)$ for all $i$.*

This lemma establishes that if $n + 1$ subspaces are sufficiently far in terms of chordal distance, then for any $n$ points there is a subspace whose $\gamma$-sector does not contain any of the $n$ points. The proof is given in Appendix D.3.3

**Lemma D.5.** *Consider $(n + 1)$ subspaces $A_1, ..., A_{n+1}$ of dimension $m$ s.t. for any $i \neq j$,*

$$\max_{x \in A_i, z \in A_j} \frac{x^T z}{\|x\|_2 \|z\|_2} < g(\gamma).$$

*Given $n$ vectors $y_1, ..., y_n \in \mathcal{B}$ (w.l.o.g. all unit norm), then there is a subspace $H \in \{A_1, ..., A_{n+1}\}$ s.t. for all $y \in \{y_1, ..., y_n\}$,*

$$\max_{w \in H} \frac{y^T w}{\|w\|_2} \leqslant \gamma,$$

*meaning $y_i \notin \overset{\triangle}{\mathcal{C}}_\gamma(H)$ for $i = 1, ..., n$.*

This lemma is similar to Lemma D.4 but uses a more explicit notion of distance between subspaces. The proof is given in Appendix D.3.4.

### D.3.2 Proof of Lemma D.3

To prove Lemma D.3, we show the existence of a subspace packing $\mathcal{C}$ in $\mathcal{G}_{m,d}(\mathbb{R})$ (with $m = 2^{\lceil N/4 \rceil}$) and use Lemma D.4. The subspace packing must satisfy two conditions:

- $|\mathcal{C}| \geqslant n + 1$.
- $d_{\min}(\mathcal{C}) \geqslant \sqrt{m - g(\gamma)^2}$.

To show the existence of a suitable subspace packing, we use Lemma D.2 with $k = \lfloor \frac{g(\gamma)}{m}\sqrt{d} + 1 \rfloor$ ($\leqslant d$). Lemma D.2 gives a packing $\mathcal{C}$ of size $|\mathcal{C}| \geqslant \left(\frac{d}{2}\right)^{\frac{g(\gamma)}{2m}\sqrt{d}-1} \lfloor \frac{d}{m} \rfloor$ s.t

$$d_{\min}(\mathcal{C}) \geqslant \sqrt{m - g(\gamma)^2}.$$

Substituting in $m = 2^{\lceil N/4 \rceil}$ into the lower-bound on the size of $\mathcal{C}$ gives

$$\lfloor \frac{d}{m} \rfloor \left(\frac{d}{2}\right)^{\frac{g(\gamma)\sqrt{d}}{2m}-1} = \lfloor \frac{d}{m} \rfloor \left(\frac{d}{2}\right)^{-3/4} \left(\frac{d}{2}\right)^{\frac{g(\gamma)}{2}2^{N/2-\lceil N/4 \rceil}-\frac{1}{4}} \geqslant 2^{3/4} \frac{\lfloor 2^{3N/4} \rfloor}{2^{3N/4}} \left(\frac{d}{2}\right)^{\frac{g(\gamma)}{4}2^{N/4}-\frac{1}{4}} \geqslant \left(\frac{d}{2}\right)^{\frac{g(\gamma)}{8}d^{1/4}},$$

where we used

- $N/2 - \lceil N/4 \rceil \geqslant N/4 - 1$ because $\lceil x \rceil \leqslant x + 1$.
- $2^{3/4} \frac{\lfloor 2^{3N/4} \rfloor}{2^{3N/4}} \geqslant 1$ if $N \geqslant 2$.
- $\frac{g(\gamma)}{4}2^{N/4} - \frac{1}{4} \geqslant \frac{g(\gamma)}{8}2^{N/4}$ if $N \geqslant 8$ and $\gamma \geqslant \sqrt{3/4}$.

Since the condition given in the statement of the lemma is

$$n + 1 \leqslant \left(\frac{d}{2}\right)^{\frac{g(\gamma)}{8}d^{1/4}} \leqslant |\mathcal{C}|,$$

the packing $\mathcal{C}$ satisfies $|\mathcal{C}| \geqslant n + 1$ and $d_{\min}(\mathcal{C}) \geqslant \sqrt{m - g(\gamma)^2}$. By Lemma D.4 there exists a subspace $A \in G_{m,d}(\mathbb{R})$ of dimension $m = 2^{\lceil N/4 \rceil}$ s.t.

$$\forall x \in D, \qquad x \notin \overset{\triangle}{\mathcal{C}}_\gamma(A),$$

which concludes the proof of the lemma. $\qquad\square$

### D.3.3 PROOF OF LEMMA D.4

Consider distinct $A, B \in \mathcal{C}$, i.e $A, B$ are subspaces of dimension $m$ s.t.

$$d_C(A, B) > \sqrt{m - g(\gamma)^2}.$$

Now letting $0 \leqslant \theta_1 \leqslant ... \leqslant \theta_m \leqslant \pi/2$ be the principal angles between $A$ and $B$ (see Appendix D), we have

$$d_C(H, A) = \sqrt{\sin^2(\theta_1) + ... + \sin^2(\theta_m)} \leqslant \sqrt{m - 1 + \sin^2 \theta_1}.$$

Combining with the inequality above,

$$\sqrt{m - g(\gamma)^2} \leqslant \sqrt{m - 1 + \sin^2 \theta_1} \implies \sin^2(\theta_1) > 1 - g(\gamma)^2.$$

The first principal angle $\theta_1 = \arccos a^T b$ where $a \in A$, $b \in B$ are unit vectors chosen s.t.

$$a^T b = \max_{x \in A, y \in B} \frac{x^T y}{\|x\|_2 \|y\|_2}.$$

So we have

$$\max_{x \in A, y \in B} \frac{x^T y}{\|x\|_2 \|y\|_2} = a^T b = \cos \theta_1 = \sqrt{1 - \sin^2(\theta_1)} < g(\gamma).$$

Since this applies to arbitrary distinct $A, B \in \mathcal{C}$, there are $(n+1)$ subspaces $A_1, ..., A_{n+1} \in \mathcal{C}$ of dimension $m$ s.t. for any $i \neq j$,

$$\max_{x \in A_i, z \in A_j} \frac{x^T z}{\|x\|_2 \|z\|_2} < g(\gamma) = 2\gamma^2 - 1.$$

By Lemma D.5, there exists a subspace $H$ in $\{A_1, ..., A_{n+1}\}$ s.t. for all $y \in \{y_1, ..., y_n\}$,

$$\max_{w \in H} \frac{y^T w}{\|w\|_2} \leqslant \gamma,$$

i.e. $y_i \notin \overset{\triangle}{\mathcal{C}}_\gamma(H)$ for $i = 1, ..., n$, which concludes the proof. $\qquad \square$

### D.3.4 PROOF OF LEMMA D.5

Recall that we consider $(n+1)$ subspaces $A_1, ..., A_{n+1}$ of dimension $m$ s.t. for any $i \neq j$,

$$\max_{x \in A_i, z \in A_j} \frac{x^T z}{\|x\|_2 \|z\|_2} < g(\gamma) = 2\gamma^2 - 1. \tag{1}$$

Identify each $y_i$ to the subspace $A_{h(i)}$ that is closest to $y_i$ in terms of inner product or angle, formally

$$h(i) = \arg \max_j \max_{u \in A_j} y_i^T u / \|u\|_2.$$

Since there are $n$ vectors $y_i$, there will be at least one subspace in $\{A_1, ..., A_{n+1}\}$ that is not associated to any of the $y_i$s. Call this subspace H.

**We show that for any** $y \in \{y_i\}_{i=1}^n$, $\max_{w \in H} \frac{y^T w}{\|w\|_2} \leqslant \gamma$.

Fix $y \in \{y_i\}_{i=1}^n$ and let $A$ be its corresponding subspace $A_{h(i)}$. Let

$$w = \operatorname{argmax}_{w' \in H} \frac{y^T w'}{\|w'\|_2}.$$

$$a = \operatorname{argmax}_{a' \in A} \frac{y^T a'}{\|a'\|_2}.$$

Assume both $w$ and $a$ are of unit norm w.l.o.g. We know:

- 1. $w^T y \leqslant a^T y$ from the definition of $h(i)$.
- 2. $w^T a < g(\gamma)$ from (1).

Let $x = \frac{w+a}{\|w+a\|_2}$ be the average of $w$ and $a$, with $\|w + a\|_2^2 = w^T w + 2a^T w + a^T a = 2 + 2a^T w = 2(1 + a^T w)$.

**Claim:** $w^T y \leqslant w^T x$ (intuition: $y$ is "closer" to $a$ than $w$, so $w$ should be "closer" to average of $a$ and $w$ than to $y$).

**Proof of claim:** Assume $w^T y > w^T x$. From 1. this implies $a^T y > w^T x$. Now

$$w^T x = w^T \frac{w+a}{\|w+a\|_2} = \frac{w^T w + w^T a}{\|w+a\|_2} = \frac{1 + w^T a}{\|w+a\|_2} = \frac{a^T a + w^T a}{\|w+a\|_2} = a^T x.$$

So we have $a^T y > a^T x \implies a^T(y-x) > 0$. From the initial assumption we also have $w^T(y-x) > 0$. Combining:

$$
\begin{aligned}
(w + a)^T(y - x) > 0 &\implies x^T(y - x) > 0 \\
&\implies x^T y - x^T x > 0 \\
&\implies x^T y - 1 > 0 \\
&\implies x^T y > 1,
\end{aligned}
$$

which is a contradiction since both $x$ and $y$ are of unit-norm. **End of proof of claim.**

We now show $w^T x \leqslant \gamma$. Using 2.,

$$w^T x = \frac{1 + w^T a}{\|w+a\|_2} = \frac{\sqrt{1 + a^T w}}{\sqrt{2}} \leqslant \frac{\sqrt{1 + g(\gamma)}}{\sqrt{2}} = \frac{\sqrt{1 + 2\gamma^2 - 1}}{\sqrt{2}} = \frac{\sqrt{2\gamma^2}}{\sqrt{2}} = \gamma.$$

Combining with the claim we have that $w^T y \leqslant \gamma$. Given the definition of $w$ and that $y$ was arbitrary, we have shown

$$\arg\max_i \max_{w \in H} \frac{y_i^T w}{\|w\|_2} \leqslant \gamma,$$

which concludes the proof. □

# E  PROOF OF THEOREM 4.4

## E.1  MDP CLASS

The PE problems used in the proof of Theorem 4.4 are characterised by a class of MDPs $\mathcal{M}$ and target policies $\Pi$. In this section, we define the class of MDPs $\mathcal{M}$. All MDPs $M \in \mathcal{M}$ in the class share the same state-space $\mathcal{S}$, action space $\mathcal{A}$, feature map and transition function $p$ but differ in the reward function $r_M$ and target policy $\pi_M$. This class of MDPs is the same as the one used by Zanette (2021) in the proof of their Theorem 1. Our constructions differ in the set of target policies $\Pi_M \in \Pi$ which are defined in Appendix E.2.

- **State-space:** $\mathcal{S} = \mathcal{B}$ and the starting state is the origin $\bar{s} = \mathbf{0} \in \mathcal{B}$.
- **Action-space:** For all $s \in \mathcal{S}$, $\mathcal{A}_s = \mathcal{A} = \mathcal{B}$.
- **Feature-map:** The feature map $\phi$ maps a state-action pair $(s, a)$ to the action $a$, i.e. $\forall (s, a), \phi(s, a) = a$. Since $a \in \mathcal{B}$, the feature space is the unit-hypersphere $\mathcal{B}$ and $\|\phi(\cdot, \cdot)\|_2 \leqslant 1$ holds for all inputs.
- **Transition-function:** The successor state of a state-action pair $(s, a)$ is deterministic and is the action $a$. This is well-defined because both the action and state space are $\mathcal{B}$. This only depends on the chosen action (and not the current state), so we will denote the unique successor state when taking action $a$ by $s^+(a) = a$.

The MDP class $\mathcal{M}$ is known to the learner (see Section 3). Therefore the learner knows the transition function and that all MDPs share it. Therefore only the feedback of the reward function and target policy is useful to the learner, as both of these are unknown. We define them in the following section.

## E.2  INSTANCE OF THE CLASS

Every MDP $M \in \mathcal{M}$ is fully characterised by a vector $w \in \partial \mathcal{B}$ and a sign $+$ or $-$. Hence, they are denoted by $M_{w,+}$ or $M_{w,-}$ and the **reward function** associated to either MDP will be denoted with the same subscript. Specifically it is defined as follows:

$$\text{on } M_{w,+}: \qquad r_{w,+}(s, a) = \begin{cases} 0, & \text{if } a \notin \mathcal{C}_\gamma(w) \cup \mathcal{C}_\gamma(-w). \\ +(1-\gamma)a^T w, & \text{otherwise.} \end{cases}$$

$$\text{on } M_{w,-}: \qquad r_{w,-}(s, a) = \begin{cases} 0, & \text{if } a \notin \mathcal{C}_\gamma(w) \cup \mathcal{C}_\gamma(-w). \\ -(1-\gamma)a^T w, & \text{otherwise.} \end{cases}$$

See Appendix C for the definition of hyper-spherical caps $\mathcal{C}_\gamma(x)$. Note that the transition function is the same for all MDPs in the class $\mathcal{M}$ and is defined in Appendix E.1. In particular, the two MDPs $M_{w,+}$ or $M_{w,-}$ only differ in their reward functions, which are opposite.

**Target policy:** The set of target policies $\Pi_M$ for an MDP $M \in \mathcal{M}$ depends on the vector $w \in \partial \mathcal{B}$ that partially characterises the MDP but not on the sign $\pm$. The set of target policies is therefore the same for $M_{w,+}$ and $M_{w,-}$.

Fix $K > 0$ and consider a sequence of $K$ nested (not necessarily strictly) subspaces of $\mathbb{R}^d$:

$$B_K \subset B_{K-1} \subset ... \subset B_2 \subset B_1 \subset B_0 = \mathbb{R}^d,$$

s.t. $\dim B_K > 0$ and $w \in B_K$. Set $B_{K+1} = \langle w \rangle$. Let $H_w = \{B_1, ..., B_K, B_{K+1}\}$ denote the set of nested subspaces (including $\langle w \rangle$). This is not defined as an ordered set, but for notational purposes the order can always be recovered since the sequence must be nested. If $A$ is a subspace of $\mathbb{R}^d$, let $\text{proj}_A(x)$ denote the orthogonal projection of $x$ onto $A$. See Appendix C for the definition of hyper-spherical caps $\mathcal{C}_\gamma(w)$ and sectors $\overset{\triangle}{\mathcal{C}}_\gamma(H)$. A target policy $\pi_{H_w}$ is specified by the set of nested subspaces $H_w$ and is defined as:

$$\pi_{H_w}(s) = \begin{cases} \frac{1}{\gamma} \text{proj}_{B_{k+1}}(s), & \text{if } s \in \overset{\triangle}{\mathcal{C}}_\gamma(B_k) \backslash \overset{\triangle}{\mathcal{C}}_\gamma(B_{k+1}) \text{ for } k = 0, ..., K. \\ \frac{1}{\gamma} \text{proj}_{B_{K+1}}(s), & \text{if } s \in \overset{\triangle}{\mathcal{C}}_\gamma(B_{K+1}) \backslash (\mathcal{C}_\gamma(w) \bigcup \mathcal{C}_\gamma(-w)). \\ s, & \text{if } s \in \mathcal{C}_\gamma(w) \bigcup \mathcal{C}_\gamma(-w). \end{cases}$$

The target policy is defined for all $s \in \mathcal{S} = \mathcal{B}$:

$$\Big[ \bigcup_{k=0}^{K} \overset{\triangle}{\mathcal{C}}_{\gamma}(B_k) \backslash \overset{\triangle}{\mathcal{C}}_{\gamma}(B_{k+1}) \Big] \bigcup \Big[ \overset{\triangle}{\mathcal{C}}_{\gamma}(B_{K+1}) \backslash (\mathcal{C}_{\gamma}(w) \bigcup \mathcal{C}_{\gamma}(-w)) \Big] \bigcup \Big[ \mathcal{C}_{\gamma}(w) \bigcup \mathcal{C}_{\gamma}(-w) \Big] = \overset{\triangle}{\mathcal{C}}_{\gamma}(B_0)$$

$$= B_0$$
$$= \mathcal{B}.$$

and the pair-wise intersections are all empty (so they form a partition). The actions taken are also well-defined:

- For $k = 0, ..., K$, if $s \in \overset{\triangle}{\mathcal{C}}_{\gamma}(B_k) \backslash \overset{\triangle}{\mathcal{C}}_{\gamma}(B_{k+1})$, then $s \notin \overset{\triangle}{\mathcal{C}}_{\gamma}(B_{k+1})$ and from Definition C.2, this means that for any $u \in B_{k+1}$, $|u^T s| / \|u\|_2 \|s\|_2 \leqslant \gamma$. Denoting $x = \text{proj}_{B_{k+1}}(s) \in B_{k+1}$,

$$\|x\|_2 = \frac{x^T x}{\|x\|_2} = \frac{|x^T s|}{\|x\|_2} \leqslant \gamma \|s\|_2 \leqslant \gamma,$$

  meaning that $\frac{1}{\gamma} \text{proj}_{B_{k+1}}(s) \in \mathcal{B}$.

- If $s \in \overset{\triangle}{\mathcal{C}}_{\gamma}(B_{K+1}) \backslash (\mathcal{C}_{\gamma}(w) \cup \mathcal{C}_{\gamma}(-w))$, , then $s \notin (\mathcal{C}_{\gamma}(w) \cup \mathcal{C}_{\gamma}(-w))$ and from Definition C.1, this means that $|w^T s| \leqslant \gamma$. Denoting $x = \text{proj}_{B_{K+1}}(s) \in B_{K+1}$,

$$\|x\|_2 = \frac{x^T x}{\|x\|_2} = \frac{|x^T s|}{\|x\|_2} \leqslant \gamma,$$

  meaning that $\frac{1}{\gamma} \text{proj}_{B_{K+1}}(s) \in \mathcal{B}$.

The **set of target policies** $\Pi_{M_{w,\pm}}$ for the MDP $M_{w,\pm}$ is the set of (deterministic) policies $\pi_{H_w}$ for any sequence of nested subspaces $H_w$ satisfying the above conditions ($w \in B_K$). We sometimes use a $w, \pm$ subscript to refer to both MDPs simultaneously.

Crucially observing actions in $\overset{\triangle}{\mathcal{C}}_{\gamma}(B_k) \backslash \overset{\triangle}{\mathcal{C}}_{\gamma}(B_{k+1})$ for $k \leqslant K$ may not reveal $w$. Without the knowledge of $w$, the reward function is unknown and even with the knowledge of $w$ or the target policy the reward function is not fully identified, in which case $M_{w,+}$ and $M_{w,-}$ cannot be distinguished.

### E.3 REALIZABILITY:

We show that the action-value of any target policy for any MDP $M \in \mathcal{M}$ can be linearly represented with the feature map defined in Appendix E.1. The action-value of a target policy $\pi_{H_w}$ only depends on $w$ and the sign $\pm$ of the reward function of the MDP, the sequence of nested subspace does not matter beyond $w$.

**Lemma E.1** ($Q^\pi$ Realizability). *For any $w \in \partial \mathcal{B}$ let $Q_{w,+}$ and $Q_{w,-}$ be the action-value functions of any $\pi \in \Pi_{M_{w,\pm}}$ (a target policy) on $M_{w,+}$ and $M_{w,-}$, respectively. Then it holds that*

$$\forall (s,a), \qquad \begin{cases} Q_{w,+}(s,a) = +\phi(s,a)^T w & \text{on } M_{w,+}. \\ Q_{w,-}(s,a) = -\phi(s,a)^T w & \text{on } M_{w,-}. \end{cases}$$

**Proof:** Consider $M_{w,+}$ and set $Q(s,a) = \phi(s,a)^T w$. We will show that $Q$ satisfies the Bellman evaluation equations for $\pi$ at all state-actions pairs, which will imply that $Q(s,a) = Q_{w,+}(s,a)$. Apply $\mathcal{T}^\pi_{M_{w,+}}$ to $Q$ at $(s,a)$:

$$\mathcal{T}^\pi_{M_{w,+}}(Q)(s,a) = r_{w,+}(s,a) + \gamma Q(s^+(a), \pi(s^+(a)))$$
$$= r_{w,+}(s,a) + \gamma Q(a, \pi(a)),$$

since $s^+(a) = a$ is the successor state of $(s,a)$. We consider the RHS of the above for all possible cases.

**Case 1:** If $a \in \overset{\triangle}{\mathcal{C}}_\gamma(B_k) \backslash \overset{\triangle}{\mathcal{C}}_\gamma(B_{k+1})$ for some $k \in [0, ..., K]$:

$$r_{w,+}(s,a) + \gamma Q(a, \pi(a)) = 0 + \gamma Q(a, \frac{1}{\gamma} \text{proj}_{B_{k+1}}(a))$$

$$= \gamma \phi(a, \frac{1}{\gamma} \text{proj}_{B_{k+1}}(a))^T w$$

$$= \gamma \frac{1}{\gamma} \text{proj}_{B_{k+1}}^T(a) w$$

$$= \text{proj}_{B_{k+1}}^T(a) w.$$

However, recall that $w \in B_{k+1}$. Since $\text{proj}_{B_{k+1}}(a)$ is the **orthogonal** projection of $a$: $w^T \text{proj}_{B_{k+1}}(a) = w^T a$. Plugging into the above, we have: $r_{w,+}(s,a) + \gamma Q(a, \pi(a)) = a^T w = \phi(s,a)^T w = Q(s,a)$, which satisfies the Bellman evaluation equation.

**Case 2:** If $a \in \overset{\triangle}{\mathcal{C}}_\gamma(B_{K+1}) \backslash (\mathcal{C}_\gamma(w) \cup \mathcal{C}_\gamma(-w))$, as above (recalling $B_{K+1} = \langle w \rangle$):

$$r_{w,+}(s,a) + \gamma Q(a, \pi(a)) = 0 + \gamma Q(a, \frac{1}{\gamma} \text{proj}_{B_{K+1}}(a))$$

$$= \gamma \phi(a, \frac{1}{\gamma} \text{proj}_{B_{K+1}}(a))^T w$$

$$= \gamma \frac{1}{\gamma} \text{proj}_{B_{K+1}}^T(a) w$$

$$= \text{proj}_{B_{K+1}}^T(a) w$$

$$= (w^T a) w^T w = w^T a = \phi(s,a)^T w = Q(s,a),$$

which satisfies the Bellman evaluation equation.

**Case 3:** If $a \in \mathcal{C}_\gamma(w) \cup \mathcal{C}_\gamma(-w)$, the reward is no longer 0 and we have:

$$r_{w,+}(s,a) + \gamma Q(a, \pi(a)) = (1-\gamma) a^T w + \gamma Q(a, a)$$

$$= (1-\gamma) a^T w + \gamma \phi(a, a)^T w$$

$$= (1-\gamma) a^T w + \gamma a^T w$$

$$= a^T w = \phi(s,a)^T w = Q(s,a),$$

which satisfies the Bellman evaluation equation.

For all cases, $Q$ satisfies the Bellman evaluation equations, so it is the fixed point of the Bellman evaluation operator. In particular, it is the action-value of the target policy $\pi$ on $M_{w,+}$.

Now consider $M_{w,-}$ and set $Q(s,a) = -\phi(s,a)^T w$. We will show that $Q$ satisfies the Bellman evaluation equations for $\pi$ at all state-actions pairs, which will imply that $Q(s,a) = Q_{w,-}(s,a)$. As above, apply $\mathcal{T}_{M_{w,-}}^\pi$ at $(s,a)$ to $Q$:

$$\mathcal{T}_{M_{w,-}}^\pi(Q)(s,a) = r_{w,-}(s,a) + \gamma Q(a, \pi(a)),$$

and consider the RHS of the above for all possible cases.

**Case 1:** If $a \in \overset{\triangle}{\mathcal{C}}_\gamma(B_k) \backslash \overset{\triangle}{\mathcal{C}}_\gamma(B_{k+1})$ for some $k \in [0, ..., K]$:

$$r_{w,-}(s,a) + \gamma Q(a, \pi(a)) = 0 + \gamma Q(a, \frac{1}{\gamma} \text{proj}_{B_{k+1}}(a))$$

$$= -\gamma \phi(a, \frac{1}{\gamma} \text{proj}_{B_{k+1}}(a))^T w$$

$$= -\gamma \frac{1}{\gamma} \text{proj}_{B_{k+1}}^T(a) w$$

$$= -\text{proj}_{B_{k+1}}^T(a) w$$

$$= -a^T w = -\phi(s,a)^T w = Q(s,a).$$

**Case 2:** If $a \in \overset{\triangle}{\mathcal{C}}_\gamma(B_{K+1})\backslash(\mathcal{C}_\gamma(w) \cup \mathcal{C}_\gamma(-w))$, as above:

$$
\begin{aligned}
r_{w,-}(s,a) + \gamma Q(a, \pi(a)) &= 0 + \gamma Q(a, \frac{1}{\gamma}\mathrm{proj}_{B_{K+1}}(a)) \\
&= -\gamma \phi(a, \frac{1}{\gamma}\mathrm{proj}_{B_{K+1}}(a))^T w \\
&= -\gamma \frac{1}{\gamma}\mathrm{proj}_{B_{K+1}}^T(a) w \\
&= -\mathrm{proj}_{B_{K+1}}^T(a) w \\
&= -(w^T a)w^T w = -w^T a = -\phi(s,a)^T w = Q(s,a).
\end{aligned}
$$

**Case 3:** If $a \in \mathcal{C}_\gamma(w) \cup \mathcal{C}_\gamma(-w)$, the reward is no longer 0 and we have:

$$
\begin{aligned}
r_{w,-}(s,a) + \gamma Q(a, \pi(a)) &= -(1-\gamma)a^T w + \gamma Q(a,a) \\
&= -(1-\gamma)a^T w - \gamma \phi(a,a)^T w \\
&= -(1-\gamma)a^T w - \gamma a^T w \\
&= -a^T w = -\phi(s,a)^T w = Q(s,a),
\end{aligned}
$$

which satisfies the Bellman evaluation equation.

For all cases, $Q$ satisfies the Bellman evaluation equations, so it is the fixed point of the Bellman evaluation operator. In particular, it is the action-value of the target policy $\pi$ on $M_{w,-}$. $\qquad\square$

### E.4 PROOF OF THEOREM 4.4

Consider the MDP class described in Appendix E.1 and Appendix E.2. First, we know from Lemma E.1 ($Q^\pi$ Realizability) that all instances of the PE problem characterised by the MDP class $\mathcal{M}$ and target policies $\Pi$ satisfy Assumption 4.1 ($Q^\pi$ is Realizable) with the feature map $\phi(\cdot, \cdot)$ defined in Appendix E.1.

The dynamics of the MDPs in the class $\mathcal{M}$ are the same, which as discussed in Remark 3.6 means that policy-induced and policy-free queries are equivalent. We provide a proof for policy-free queries.

We specify the instance with MDP $M \in \mathcal{M}$ and target policy $\pi_M \in \Pi_M$ according to the queries selected by the learner. Fix $K > 0$. Recall that $n_k$ is the number of queries made by the learner at round $k$ and $n = \sum_{k=1}^K n_k$ is the total number of queries. Let $A_k$ be the set of actions queried by the learner at round k (i.e. $|A_k| = n_k$). Set $\bar{A}_k = \bigcup_{i=1}^k A_k$ to be the set of all actions queried up to round $k$.

The learner is sample-efficient so $n$ is polynomial in $d$. Specifically, there exists some constant $\alpha > 0$ and some integer $T$ such that $n \leqslant \alpha d^T$. Consider $N \in \mathbb{N}$ s.t. $2^N \leqslant d < 2^{N+1}$ and set $d_+ = 2^N$. Note that $d_+ \geqslant d/2$. Fix $W = \exp(\frac{1}{8}g(\gamma)d_+^{1/4^K})$. If $n \leqslant W$, we can show the learner cannot solve the PE problem. Consider both cases:

### E.4.1   CASE 1

Suppose $W < n$, then

$$
W < n \implies W < \alpha d^T
$$
$$
\implies \frac{g(\gamma)}{8} d_+^{1/4^K} < \log(\alpha d^T)
$$
$$
\implies \frac{g(\gamma)}{8} (d/2)^{1/4^K} < \log(\alpha d^T) \qquad \text{using that } d_+ \geqslant \frac{d}{2}
$$
$$
\implies (d/2)^{1/4^K} < \frac{8}{g(\gamma)} \log(\alpha d^T)
$$
$$
\implies \frac{1}{4^K} \log(d/2) < \log\left(\frac{8}{g(\gamma)} \log(\alpha d^T)\right)
$$
$$
\implies \frac{\log(d/2)}{\log\left(\frac{8}{g(\gamma)} \log(\alpha d^T)\right)} < 4^K
$$
$$
\implies \frac{1}{\log 4} \log\left[\frac{\log(d/2)}{\log\left(\frac{8}{g(\gamma)} \log(\alpha d^T)\right)}\right] < K
$$
$$
\implies K \geqslant c' \log\log d \qquad \text{for a constant } c' > 0 \text{ and } d \text{ sufficiently large.}
$$

### E.4.2   CASE 2

$n \leqslant W \implies n_k \leqslant \exp(\frac{1}{8} g(\gamma) d_+^{1/4^k})$ for $k = 1, ..., K$. We inductively define the sequence of nested subspaces $B_K \subset ... \subset B_1$ characterising $w \ (\in B_K)$ and used by the target policy $\pi_M$ s.t. for $k = 1, ..., K$, $\dim B_k = 2^{\lceil N/4^k \rceil} \geqslant 2^8$ and

$$
\forall a \in \bar{A}_k, \qquad a \notin \overset{\triangle}{\mathcal{C}}_\gamma(B_k).
$$

**Proof of existence of nested subspaces:**

We proceed by induction. Let $B_K \subset ... \subset B_1$ be an arbitrary sequence of subspaces and $w \in B_K \cap \mathcal{B}$ arbitrary. We will define these but for now consider the target policy $\pi_{H_w}$ (see Appendix E.2) with $H_w = \{\langle w \rangle, B_K, ..., B_1\}$.

**Base Case:** At round $k = 1$, the learner has observed no feedback and chooses a set of action queries $A_1 = \bar{A}_1$ (we ignore the queried states since the reward and transitions functions only depend on the action) such that:

$$
|A_1| = n_1 \leqslant \exp(\frac{1}{8} g(\gamma) d_+^{1/4}) \leqslant \left(\frac{d_+}{2}\right)^{\frac{1}{8} g(\gamma) d_+^{1/4}} - 1 \qquad (\text{if } d_+ \geqslant 12),
$$

so by Lemma D.3 there exists a subspace $H \in \mathcal{G}_{2^{\lceil N/4 \rceil}, d_+}(\mathbb{R})$ of dimension $2^{\lceil N/4 \rceil} \ (\geqslant d_+^{1/4})$ such that

$$
\forall x \in A_1 = \bar{A}_1, \qquad x \notin \overset{\triangle}{\mathcal{C}}_\gamma(H).
$$

Set $B_1 = H$. The learner then observes the feedback for $A_1$ (The state $s$ in the reward does not matter):

$$
\{(r_{w,\pm}(s,a), \pi_{H_w}(s^+(a))) : a \in A_1\}.
$$

Not having fixed $B_2, ..., B_K, w$ does not cause problems with the feedback even though $\pi_{H_w}$ and $r_{w,\pm}$ depend on them since the feedback of $A_1$ only depends on $B_1$: the learner only observes the projection of the actions in $A_1$ into $B_1$,

$$
\forall a \in \bar{A}_1, \qquad a \notin \overset{\triangle}{\mathcal{C}}_\gamma(B_1)
$$
$$
\implies \forall a \in \bar{A}_1, \qquad \pi_{H_w}(s^+(a)) = \pi_{H_w}(a) = \frac{1}{\gamma} \text{proj}_{B_1}(a), \qquad r_{w,\pm}(s,a) = 0.
$$

In particular, there is no dependence on $w$ or $B_{1+i} \subset B_1$ for $i \geqslant 1$, which can be arbitrary and fixed in later rounds.

**Inductive Step:** Suppose that at round $k$, there exists a sequence of nested subspaces $B_k \subset ... \subset B_1$ used by the target policy $\pi_{H_w}$ s.t. for $i = 1, ..., k$, $\dim B_i = 2^{\lceil N/4^i \rceil} \geqslant 2^8$ and

$$\forall a \in \bar{A}_i, \qquad a \notin \overset{\triangle}{\mathcal{C}}_\gamma(B_i).$$

$B_{k+1}$ need not be fixed yet since the feedback of $\bar{A}_k$ only depends on $B_1, ..., B_k$:

$$\forall a \in A_k, \qquad a \notin \overset{\triangle}{\mathcal{C}}_\gamma(B_k)$$
$$\implies \forall a \in A_k, \qquad \pi_{H_w}(s^+(a)) = \pi_{H_w}(a) = \frac{1}{\gamma}\text{proj}_{B_j}(a) \qquad \text{for some } j \leqslant k, \qquad r_{w,\pm}(s,a) = 0.$$

The learner only observes the projection of actions in $\bar{A}_k$ into $B_1, B_2, ..., B_k$. Note that the only constraint on $M$ up to this step is $w \in B_k$.

Now $B_k$ is such that $\forall x \in \bar{A}_k$, $x \notin \overset{\triangle}{\mathcal{C}}_\gamma(B_k)$ with $d_k = \dim B_k = 2^{\lceil N/4^k \rceil} \geqslant d_+^{1/4^k} \geqslant 2^8$. The learner has observed the feedback from $\bar{A}_k$ and chooses a set of action queries $A_{k+1}$ such that

$$|A_{k+1}| = n_{k+1} \leqslant \exp(\frac{1}{8}g(\gamma)d_+^{1/4^{k+1}}) \leqslant \left(\frac{d_k}{2}\right)^{\frac{1}{8}g(\gamma)d_+^{1/4^{k+1}}} - 1 \leqslant \left(\frac{d_k}{2}\right)^{\frac{1}{8}g(\gamma)d_k^{1/4}} - 1 \qquad \text{(if } d_+ \geqslant 12) .$$

Then we know that the number of those queries that are in $B_k$ is also less than $\left(\frac{d_k}{2}\right)^{\frac{1}{8}g(\gamma)d_k^{1/4}}$. Therefore by Lemma D.3, there exists a subspace $H \subset B_k$ of dimension $2^{\lceil \lceil N/4^k \rceil/4 \rceil} \geqslant d_{k+1} = 2^{\lceil N/4^{k+1} \rceil}$ (can reduce the dimension if they do not match, removing dimensions will not add queries to $\overset{\triangle}{\mathcal{C}}_\gamma(H)$) such that

$$\forall x \in A_{k+1}, \qquad x \notin \overset{\triangle}{\mathcal{C}}_\gamma(H).$$

In particular, $A_{k+1}$ can be entirely contained in $B_k$ or can depend on $B_k, ..., B_1$ in any arbitrary way. Set $B_{k+1} = H$. We know that $\forall x \in \bar{A}_k, x \notin \overset{\triangle}{\mathcal{C}}_\gamma(B_k)$ and $B_{k+1} \subset B_k$ so we have

$$\forall x \in \bar{A}_{k+1}, \qquad x \notin \overset{\triangle}{\mathcal{C}}_\gamma(B_{k+1}).$$

The learner then observes the feedback for $A_{k+1}$ (The state $s$ in the reward does not matter):

$$\{(r_{w,\pm}(s,a), \pi_{H_w}(s^+(a))) : a \in A_{k+1}\}.$$

Not having fixed $B_{k+2}, ..., B_K, w$ does not cause problems with the feedback even though $\pi_{H_w}$ and $r_{w,\pm}$ depend on them since the feedback observed up to this round of $\bar{A}_{k+1}$ only depends on $B_1, ..., B_{k+1}$: the learner only observes the projection of the actions in $\bar{A}_{k+1}$ into $B_1, ..., B_{k+1}$,

$$\forall a \in \bar{A}_{k+1}, \qquad a \notin \overset{\triangle}{\mathcal{C}}_\gamma(B_{k+1})$$
$$\implies \forall a \in \bar{A}_{k+1}, \qquad \pi_{H_w}(s^+(a)) = \pi_{H_w}(a) = \frac{1}{\gamma}\text{proj}_{B_j}(a) \qquad \text{for some } j \leqslant k+1, \qquad r_{w,\pm}(s,a) = 0.$$

In particular, there is no dependence on $w$ or $B_{k+1+i} \subset B_{k+1}$ for $i \geqslant 1$, which can be arbitrary and fixed in later rounds.

We remark Lemma D.3 establishes the existence of a subspace in $\mathcal{G}_{2^{\lceil \lceil N/4^k \rceil/4 \rceil}, d_k}(\mathbb{R})$ where the ambient space is $\mathbb{R}^{d_k}$ instead of $\mathbb{R}^d$. Any $d_k$-dimensional subspace of $\mathbb{R}^d$ is isomorphic to $\mathbb{R}^{d_k}$, so we can consider $B_k$, project the points of $A_{k+1}$ into $B_k$, apply Lemma D.3 to get $H$ within $B_k$, and extend $H$ to be defined in $\mathbb{R}^d$. We omit these steps here for clarity but detail them fully in Appendix E.4.3. This also holds for the base case (where $d_+$ is used as the ambient dimension instead of $d$)

When applying Lemma D.3 with ambient dimension $d_k = 2^{\lceil N/4^k \rceil}$, we require $\lceil N/4^k \rceil \geqslant 8$. Since $d_k \geqslant d_+^{1/4^k}$ and $d_+ \geqslant d/2$, it is enough to have

$$(d/2)^{1/4^k} \geqslant 2^8 \iff K \leqslant \frac{1}{\log 4} \log \left( \frac{1}{8 \log 2} \log \frac{d}{2} \right)$$
$$\impliedby K \leqslant c \log \log d,$$

for a constant $c > 0$ and $d$ sufficiently large. If this condition on $K$ does not hold, then

$$K > c \log \log d. \tag{2}$$

Suppose it does hold. Then the steps above go through and we have the existence of our nested subspaces. We also have $d_k \geqslant 2^8 > 12$ for all $k = 1, ..., K$, which was needed in some of the steps.

**End of proof of existence of nested subspaces.**

We now fully specify $M$. The complete set of queried actions by the learner is $\bar{A}_K$, and

$$\forall a \in \bar{A}_K, \qquad a \notin \overset{\triangle}{\mathcal{C}}_\gamma(B_K).$$

Since $\dim B_K \geqslant 2^8 > 0$, we can pick some $w \in B_K \cap \partial \mathcal{B}$ and let $H_w = \{B_1, ..., B_K, B_{K+1}\}$ with $B_{K+1} = \langle w \rangle$. From the proof of the existence of nested spaces, we can consider the PE problem instances with MDPs $M_{w,+}$ and $M_{w,-}$ and $\pi_{H_w}$ as target policy. The transition function and target policy are the same on $M_{w,+}$ and $M_{w,-}$. Since $\forall a \in \bar{A}_K$, $a \notin \mathcal{C}_\gamma(w) \cup \mathcal{C}_\gamma(-w)$ because $a \notin \overset{\triangle}{\mathcal{C}}_\gamma(B_K)$, the reward function for any query $a \in \bar{A}_K$ is 0. Thus, the learner cannot distinguish $M_{w,+}$ between $M_{w,-}$ from the submitted queries.

The learner has to produce an estimate of $Q^{\pi_{H_w}}(\bar{s}, \cdot)$ for $\bar{s} = \mathbf{0}$. But from Lemma E.1,

$$Q^{\pi_{H_w}}(\bar{s}, w) = \phi(\bar{s}, w)^T w = w^T w = 1 \qquad \text{on } M_{w,+}.$$
$$Q^{\pi_{H_w}}(\bar{s}, w) = -\phi(\bar{s}, w)^T w = -w^T w = -1 \quad \text{on } M_{w,-}.$$

If the learner predicts a positive value for $Q^{\pi_{H_w}}(\bar{s}, w)$, it will incur an error greater than 1 on $M_{w,-}$ and similarly for $M_{w,+}$ if it predicts a negative value. Even if it randomizes between both, with probability at least $1/2$ it will incur an error of at least 1 on one of the MDPs. Therefore, the learner can be at most $(1, 1/2)$-sound.

Therefore, to be more than $(1, 1/2)$-sound, we must either be in Case 1 or be in Case 2 and satisfy condition (2). In either case, we have the condition

$$K > c \log \log d,$$

for some constant $c > 0$ and $d$ sufficiently large, which gives $K = \Omega(\log \log d)$, showing the result. $\qquad \square$

### E.4.3 DEALING WITH SWITCHES IN AMBIENT SPACE

In this section we write out the missing details of the proof of Theorem 4.4 in Appendix E.4. In particular, assuming the condition on $n_{k+1}$ is satisfied, we show that we can use Lemma D.3 for the existence of a subspace $H \subset B_k \subset \mathbb{R}^d$ of dimension $d_{k+1}$ s.t.

$$\forall x \in A_{k+1}, \qquad x \notin \overset{\triangle}{\mathcal{C}}_\gamma(H). \tag{3}$$

Fix $k \in \{0, 1, ..., K-1\}$ and set $d_0 = d_+$ and $B_0$ to any $d_+$-dimensional subspace of $\mathbb{R}^d$. Picking up the proof of in Appendix E.4, the condition on $n_{k+1} = |A_{k+1}|$ for Lemma D.3 is satisfied. $B_k \in \mathcal{G}_{d_k,d}(\mathbb{R})$ is a $d_k$-dimensional subspace within $\mathbb{R}^d$. We find an orthonormal basis for $B_k$:

$$\exists v_1, ..., v_{d_k} \in \mathbb{R}^d \text{ s.t. } B_k = \langle v_1, ..., v_{d_k} \rangle \text{ and } v_i^T v_i = 1, v_i^T v_j = 0 \text{ for } i \neq j.$$

Any vector in $B_k$ can be written as $\sum_{m=1}^{d_k} \alpha_m v_m$ for some $(\alpha_m)_{m=1}^{d_k}$. $B_k$ is isomorphic to $\mathbb{R}^{d_k}$ through the linear transformation $T : B_k \to \mathbb{R}^{d_k}$ (which can be shown to be a bijection) defined as

$$T\left(\sum_{m=1}^{d_k} \alpha_m v_m\right) = [\alpha_1, ..., \alpha_{d_k}]^T \in \mathbb{R}^{d_k}.$$

Letting $proj_{B_k}(x)$ denote the orthogonal projection of $x \in \mathbb{R}^d$ onto $B_k$, consider

$$A_{k+1}^p = \{T(proj_{B_k}(x)) : x \in A_{k+1}\} \subset \mathbb{R}^{d_k}.$$

The size of $A_{k+1}^p$ is $|A_{k+1}^p| \leqslant |A_{k+1}| = n_{k+1}$ so we can apply Lemma D.3: there exists a subspace $H^p \in \mathcal{G}_{d_{k+1},d_k}(\mathbb{R})$ s.t.

$$\forall x^p \in A_{k+1}^p, \qquad x \notin \overset{\triangle}{\mathcal{C}}_\gamma(H^p).$$

Recall $d_k = 2^{\lceil N/4^k \rceil}$ and the lemma gives a subspace of dimension $2^{\lceil \lceil N/4^k \rceil /4 \rceil} \geqslant d_{k+1} = 2^{\lceil N/4^{k+1} \rceil}$ but we can reduce the dimension if they do not match.

$H^p \in \mathcal{G}_{d_{k+1},d_k}(\mathbb{R})$ is a $d_{k+1}$-dimensional subspace within $\mathbb{R}^{d_k}$. We find an orthonormal basis for $H^p$:

$$\exists u_1^p, ..., u_{d_{k+1}^p} \in \mathbb{R}^{d_k} \text{ s.t. } H^p = \langle u_1^p, ..., u_{d_{k+1}}^p \rangle \text{ and } (u_i^p)^T u_i^p = 1, (u_i^p)^T u_j^p = 0 \text{ for } i \neq j.$$

Define

$$H = \langle T^{-1}(u_1^p), ..., T^{-1}(u_{d_{k+1}}^p) \rangle.$$

Remark that $T^{-1}(u_i^p) = \sum_{m=1}^{d_k} u_i^p(m) v_m \in B_k \subset \mathbb{R}^d$ so $H \subset B_k$. We use the notation $x(m)$ for a vector $x$ to refer to the $m$-th coordinate of $x$. Since $\{v_1, ..., v_{d_k}\}$ is an orthonormal set:

$$(T^{-1}(u_i^p))^T T^{-1}(u_j^p) = \sum_{m=1}^{d_k} u_i^p(m) u_j^p(m) = (u_i^p)^T u_j^p = \begin{cases} 1, & \text{if } i = j. \\ 0, & \text{if } i \neq j. \end{cases}$$

So $\{T^{-1}(u_1^p), ..., T^{-1}(u_{d_{k+1}}^p)\}$ is an orthonormal basis for $H$, which means $H \in \mathcal{G}_{d_{k+1},d}(\mathbb{R})$ is a $d_{k+1}$ dimensional subspace within $\mathbb{R}^d$. $H$ is the subspace we are trying to show the existence of, it remains to verify the condition (3). The following claim concludes this section.

**Claim:** $\forall x \in A_{k+1}, x \notin \overset{\triangle}{\mathcal{C}}_\gamma(H).$

**Proof of Claim:** Suppose not: $\exists x \in A_{k+1}$ s.t. $x \in \overset{\triangle}{\mathcal{C}}_\gamma(H)$. Then there exists a unit-normed $h \in H \subset B_k$ such that $x^T h > \gamma$. Since $h \in B_k$, we also have $proj_{B_k}(x)^T h > \gamma$. $h, proj_{B_k}(x)$ are both in $B_k$ so

$$\exists \alpha_1, ..., \alpha_{d_k} \in \mathbb{R} \text{ s.t. } h = \sum_{m=1}^{d_k} \alpha_m v_m,$$

$$\exists \beta_1, ..., \beta_{d_k} \in \mathbb{R} \text{ s.t. } proj_{B_k}(x) = \sum_{m=1}^{d_k} \beta_m v_m.$$

and we have

$$
\begin{aligned}
proj_{B_k}(x)^T h > \gamma \implies & \sum_{m=1}^{d_k} \alpha_m \beta_m > \gamma \\
\implies & T(proj_{B_k}(x))^T T(h) > \gamma \\
\implies & T(proj_{B_k}(x)) \in \overset{\triangle}{\mathcal{C}}_\gamma(H^p),
\end{aligned}
$$

since $T(h) \in H^p$ as $h \in H$. But $T(proj_{B_k}(x)) \in A_{k+1}^p$, which contradicts the definition of $H^p$. $\quad\square$

### E.5 ILLUSTRATION OF MDP CONSTRUCTION

In this section, we provide a high-level illustration of how the hard instance $M \in \mathcal{M}$ is constructed when $d = 3$ and the learner makes three queries per round. The reward vector of the MDP is 0 everywhere except in the $\gamma$-hyper-spherical sector $\overset{\triangle}{\mathcal{C}}_\gamma(w)$ (see Appendix C) of a vector $w \in \mathcal{B}$, which is unknown to the learner. The target policy will be constructed in such a way that $Q^\pi(s, a) = \pm\phi(s, a)^T w$. If the learner can identify $w$ and make a query in its $\gamma$ hyper-spherical sector (to distinguish between the $+$ and the $-$), then it can fully solve the PE problem. Therefore the aim of the learner in choosing its queries is to identify this vector $w$.

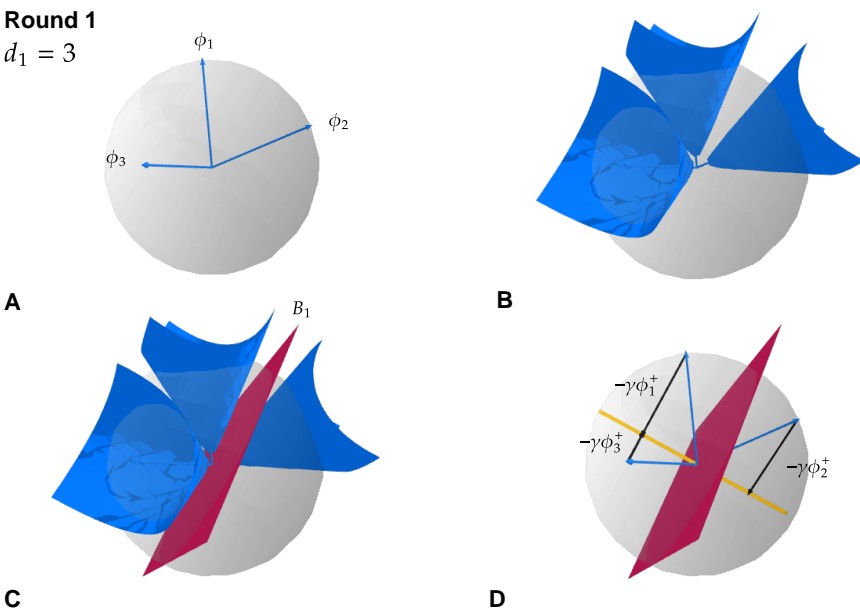

Figure 3: Round 1.

**Round 1** (Figure 3).

- A: Three queries $\phi_i = \phi(s_i, a_i) = a_i$ $(i = 1, 2, 3)$ are made by the learner.

- B: The three queries with their hyper-spherical sectors $\overset{\triangle}{\mathcal{C}}_\gamma(\phi_i)$ $(i = 1, 2, 3)$ (we only show one side of the sector for clarity).

- C: There exists a 2-dimensional subspace $B_1$ that does not intersect the hyper-spherical sectors of the queries which the environment can use to further hide $w$ - if $B_1$ does not intersect the $\gamma$ hyper-spherical sectors of the queries, then the queries $\phi_i$ will not be in the $\gamma$ hyper-spherical sector of any point in $B_1$ (in particular $w$ and cannot identify the $+$ or $-$ in the reward).

- D: The target-policy $\pi_M$ is constructed in such a way that realizability is satisfied with $w \in B_1$. In particular, $\pi_M(s_i^+) = \frac{1}{\gamma}\text{proj}_{B_1}(s_i^+) = \frac{1}{\gamma}\text{proj}_{B_1}(a_i)$ is valid because $B_1$ does not intersect the sectors of the queries (see Appendix E.2). This choice allows the component of $\phi$ in the direction of $w$ to grow by $1/\gamma$ in $\phi^+$, this will cancel with the $\gamma$ discount in the bellman equation and is a key step in satisfying realizability.

  Since the orthogonal projection is the same for all vectors in $B_1$, this definition of the target-policy (allowing realizability to be satisfied - see Appendix E.3) does not reveal any additional information about $w$ other than $w \in B_1$. In addition, $w$ or the actions taken by the policy within $B_1$ do not need to be specified yet and can depend on queries made by the learner in later rounds.

Since $\phi_i - \gamma\phi_i^+ = a_i - \mathrm{proj}_{B_1}(a_i)$ is orthogonal to $B_1$, this construction admits the interpretation of erasing information along the directions of the subspace $B_1$. Hence, this provides the intuition discussed in Section 5 but is in fact exactly what is required to hide the $w$ vector within the subspace $B_1$.

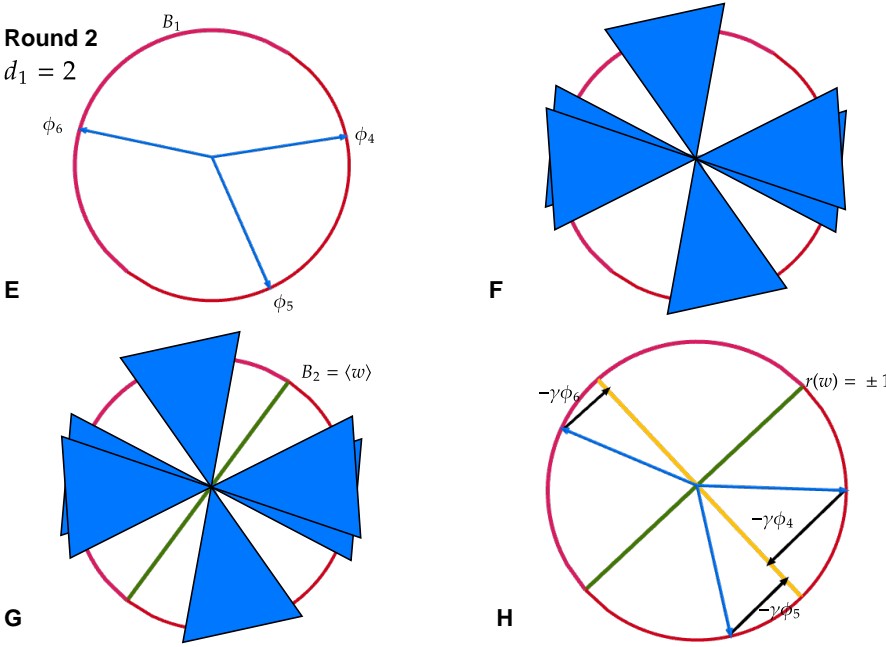

Figure 4: Round 2.

**Round 2** (Figure 4). Without loss of generality, we assume the queries made by the learner are within $B_1$ (the MDP is fully identified outside of $B_1$ so the learner can gain no further information from queries outside of $B_1$) and only visualise $B_1$ as a circle embedded in the sphere from Figure 3.

- E: Three queries $\phi_i = \phi(s_i, a_i) = a_i$ ($i = 4, 5, 6$) are made by the learner.

- F: The three queries with their (2-dimensional) hyper-spherical sectors $\overset{\triangle}{\mathcal{C}}_\gamma(\phi_i)$ ($i = 4, 5, 6$).

- G: There exists a 1-dimensional subspace $B_2 = \langle w \rangle$ that does not intersect the hyper-spherical sectors of the queries. As in C above, if $\langle w \rangle$ does not intersect the $\gamma$ hyper-spherical sectors of the queries, then the queries $\phi_i$ will not be in the $\gamma$ hyper-spherical sector of $w$.

- H: Similar to D, the target-policy $\pi_M$ is constructed within $B_1$ in such a way that realizability is satisfied. In particular, $\pi_M(s_i^+) = \frac{1}{\gamma}\mathrm{proj}_{B_2}(s_i^+) = \frac{1}{\gamma}(w^T a_i)w$ is valid because $B_2 = \langle w \rangle$ does not intersect the sectors of the queries. The reward inside $\overset{\triangle}{\mathcal{C}}_\gamma(w)$ is characterised by a sign $+$ or $-$. No queries have been made inside $\overset{\triangle}{\mathcal{C}}_\gamma(w)$ and so a third round is required to identify the sign $+$ or $-$.

This illustration uses three queries per round. In the proof, we formally show that if the number of queries in the first round is polynomial in $d$, then there always exists a subspace of dimension $d^{1/4}$ that does not intersect the sectors of the queries in the first round. We make a similar argument for later rounds to show $\log\log d$ rounds are required to identify the vector $w$. See the earlier sections for all the details.

# F    PROOF OF THEOREM 4.5

## F.1    MDP CLASS

The BPI and PE problems used in the proof of Theorem 4.5 are characterised by a class of MDPs $\mathcal{M}$ and target policies $\Pi$. In this section, we define the class of MDPs $\mathcal{M}$. All MDPs $M \in \mathcal{M}$ in the class share the same state-space $\mathcal{S}$, action space $\mathcal{A}$, feature map and target policy $\pi$ (for the PE problem) but differ in the transition function $p_M$ and reward function $r_M$. This class of MDPs is similar to the one used by Zanette (2021) in the proof of their Theorem 3. Our construction differs in the transition functions which are defined in Appendix F.2.

- **State-space:** $\mathcal{S} = \{\bar{s}\} \bigcup \mathcal{B}$ where $\bar{s}$ is the starting state disjoint from $\mathcal{B}$ (i.e. $\{\bar{s}\} \cap \mathcal{B} = \varnothing$).
- **Action-space:** Each state has a single action (which we denote by the state itself for convenience) other than the starting state $\bar{s}$ which can take actions in $\mathcal{B}$. Formally:

$$\mathcal{A}(s) = \begin{cases} \mathcal{B} & \text{if } s = \bar{s}. \\ \{s\} & \text{if } s \in \mathcal{B}. \end{cases}$$

  This notation enables that $\forall s \in \mathcal{S}, \mathcal{A}(s) \subset \mathcal{B}$.
- **Feature-map:** The feature map $\phi$ maps a state-action pair $(s, a)$ to the action $a$, i.e. $\forall (s, a), \phi(s, a) = a$. Since $a \in \mathcal{B}$, the feature space is the unit-hypersphere $\mathcal{B}$ and $\|\phi(\cdot, \cdot)\|_2 \leqslant 1$ holds for all inputs.
- **Target policy:** For the PE problem, the target policy $\pi$ is the same for all MDPs in the class: it takes action $\mathbf{0}$ in the starting state $\bar{s}$ and in the other states, there is a single action. In particular $\Pi_M = \{\pi\}$ for all $M \in \mathcal{M}$.

## F.2    INSTANCE OF THE CLASS

Fix $K > 0$ and consider a sequence of $K$ nested subspaces of $\mathbb{R}^d$:

$$B_K \subset B_{K-1} \subset ... \subset B_2 \subset B_1 \subset B_0 = \mathbb{R}^d.$$

s.t. $\dim B_K > 0$ and fix some $w \in B_K \cap \partial \mathcal{B}$. Set $B_{K+1} = \langle w \rangle$. Let $H_w = \{B_1, ..., B_K, B_{K+1}\}$ denote the set of nested subspaces (including $\langle w \rangle$). This is not defined as an ordered set, but for notational purposes the order can always be recovered since the sequence must be nested.

Every MDP $M \in \mathcal{M}$ is fully characterised by the sequence of nested subspaces $H_w$ and a sign $+$ or $-$. Hence, they are denoted by $M_{H_w,+}$ or $M_{H_w,-}$.

**Reward function:** The reward function only depends on the vector $w$ and the sign $\pm$ but we denote them with the same subscript as the MDP. Specifically it is defined as follows:

$$\text{on } M_{H_w,+}: \qquad r_{H_w,+}(s, a) = \begin{cases} 0, & \text{if } a \notin \mathcal{C}_\gamma(w) \cup \mathcal{C}_\gamma(-w). \\ +(1 - \gamma)a^T w, & \text{otherwise.} \end{cases}$$

$$\text{on } M_{H_w,-}: \qquad r_{H_w,-}(s, a) = \begin{cases} 0, & \text{if } a \notin \mathcal{C}_\gamma(w) \cup \mathcal{C}_\gamma(-w). \\ -(1 - \gamma)a^T w, & \text{otherwise.} \end{cases}$$

See Appendix C for the definition of hyper-spherical caps $\mathcal{C}_\gamma(w)$. We sometimes use a $w, \pm$ subscript to refer to both MDPs simultaneously.

**Transition Function:** The transition function for an MDP $M \in \mathcal{M}$ depends on the sequence of nested subspaces $H_w$ but not on the sign $\pm$. The transition function is therefore the same for $M_{H_w,+}$ and $M_{H_w,-}$. If $A$ is a subspace of $\mathbb{R}^d$, let $\text{proj}_A(x)$ denote the orthogonal projection of $x$ onto $A$.

See Appendix C for the definition of hyper-spherical caps $\mathcal{C}_\gamma(w)$ and sectors $\overset{\triangle}{\mathcal{C}}_\gamma(H)$. The successor state of a state-action pair $(s, a)$ is deterministic and only depends on the chosen action (and not the current state), so we will denote the unique successor state when taking action $a$ by $s_{H_w}^+(a) = a$, which is defined as:

$$s_{H_w}^+(a) = \begin{cases} \frac{1}{\gamma}\text{proj}_{B_{k+1}}(a), & \text{if } a \in \overset{\triangle}{\mathcal{C}}_\gamma(B_k) \backslash \overset{\triangle}{\mathcal{C}}_\gamma(B_{k+1}) \text{ for } k = 0, ..., K. \\ \frac{1}{\gamma}\text{proj}_{B_{K+1}}(a), & \text{if } a \in \overset{\triangle}{\mathcal{C}}_\gamma(B_{K+1}) \backslash (\mathcal{C}_\gamma(w) \cup \mathcal{C}_\gamma(-w)). \\ a, & \text{if } a \in \mathcal{C}_\gamma(w) \cup \mathcal{C}_\gamma(-w). \end{cases}$$

Note that the starting state $\bar{s}$ is not a successor state so a policy trajectory can never return to $\bar{s}$. Showing that the transition function and successor states are well defined follows the same steps as showing the target policy is well defined in the proof of Theorem 4.4 in Section E.2.

Crucially actions queried in $\overset{\triangle}{\mathcal{C}}_\gamma(B_k) \backslash \overset{\triangle}{\mathcal{C}}_\gamma(B_{k+1})$ for $k \leq K$ may not reveal $w$. Without the knowledge of $w$, the reward function is unknown and even with the knowledge of $w$ the reward function is not fully identified, in which case $M_{H_w,+}$ and $M_{H_w,-}$ cannot be distinguished.

### F.3 REALIZABILITY

We show that the action-value of any policy (not just the target policy) for any MDP $M \in \mathcal{M}$ can be linearly represented with the feature map defined in Appendix F.1. The action-value of a policy $\pi$ only depends on $w$ and the sign $\pm$ of the reward function of the MDP, the sequence of nested subspace does not matter beyond $w$.

**Lemma F.1** (Realizability). *For any $w \in \partial \mathcal{B}$ and sequence of nested subspaces $H_w$ satisfying the construction from Appendix F.2, let $Q^\pi_{H_w,+}$ and $Q^\pi_{H_w,-}$ be the action-value functions of an arbitrary policy $\pi$ on $M_{H_w,+}$ and $M_{H_w,-}$, respectively. Then it holds that*

$$\forall (s,a), \quad \begin{cases} Q^\pi_{H_w,+}(s,a) = +\phi(s,a)^T w & \text{on } M_{H_w,+}. \\ Q^\pi_{H_w,-}(s,a) = -\phi(s,a)^T w & \text{on } M_{H_w,-}. \end{cases}$$

**Proof:** Consider $M_{w,+}$ and set $Q(s,a) = \phi(s,a)^T w$. We will show that $Q$ satisfies the Bellman evaluation equations for $\pi$ at all state-actions pairs, which will imply that $Q(s,a) = Q^\pi_{H_w,+}(s,a)$. Apply $\mathcal{T}^\pi_{M_{H_w,+}}$ to $Q$ at $(s,a)$:

$$\mathcal{T}^\pi_{M_{H_w,+}}(Q)(s,a) = r_{H_w,+}(s,a) + \gamma Q(s^+_{H_w}(a), \pi(s^+_{H_w}(a))).$$

We consider the RHS of the above for all possible cases.

**Case 1:** If $a \in \overset{\triangle}{\mathcal{C}}_\gamma(B_k) \backslash \overset{\triangle}{\mathcal{C}}_\gamma(B_{k+1})$ for some $k \in [0, ..., K]$, $s^+_{H_w}(a) = \frac{1}{\gamma}\text{proj}_{B_{k+1}}(a)$. Furthermore, $\pi$ must return the only action available in the successor state (and the successor state is never $\bar{s}$), so $\pi(s^+_{H_w}(a)) = s^+_{H_w}(a)$ and we have:

$$\begin{aligned} r_{H_w,+}(s,a) + \gamma Q(s^+_{H_w}(a), \pi(s^+_{H_w}(a))) &= 0 + \gamma Q(s^+_{H_w}(a), s^+_{H_w}(a)) \\ &= \gamma \phi(s^+_{H_w}(a), s^+_{H_w}(a))^T w \\ &= \gamma s^+_{H_w}(a)^T w \\ &= \gamma \frac{1}{\gamma} \text{proj}^T_{B_{k+1}}(a) w \\ &= \text{proj}^T_{B_{k+1}}(a) w. \end{aligned}$$

However, recall that $w \in B_{k+1}$. Since $\text{proj}_{B_{k+1}}(a)$ is the **orthogonal** projection of $a$: $w^T \text{proj}_{B_{k+1}}(a) = w^T a$. Plugging into the above, we have: $\mathcal{T}^\pi_{M_{H_w,+}}(Q)(s,a) = a^T w = \phi(s,a)^T w = Q(s,a)$, which satisfies the Bellman evaluation equation.

**Case 2:** If $a \in \overset{\triangle}{\mathcal{C}}_\gamma(B_{K+1}) \backslash (\mathcal{C}_\gamma(w) \cup \mathcal{C}_\gamma(-w))$, $s^+_{H_w}(a) = \frac{1}{\gamma}\text{proj}_{B_{K+1}}(a)$, and as before the policy can only take the only action available there, giving:

$$\begin{aligned} r_{H_w,+}(s,a) + \gamma Q(s^+_{H_w}(a), \pi(s^+_{H_w}(a))) &= 0 + \gamma Q(s^+_{H_w}(a), s^+_{H_w}(a)) \\ &= \gamma \phi(s^+_{H_w}(a), s^+_{H_w}(a))^T w \\ &= \gamma s^+_{H_w}(a)^T w \\ &= \gamma \frac{1}{\gamma} \text{proj}^T_{B_{K+1}}(a) w \\ &= \text{proj}^T_{B_{K+1}}(a) w \\ &= a^T w = \phi(s,a)^T w = Q(s,a), \end{aligned}$$

which satisfies the Bellman evaluation equation.

**Case 3:** If $a \in \mathcal{C}_\gamma(w) \cup \mathcal{C}_\gamma(-w)$, the reward is no longer 0 and $s_{H_w}^+(a) = a$, and again as before the policy can only take the only action available there so we have:

$$
\begin{aligned}
r_{H_w,+}(s,a) + \gamma Q(s_{H_w}^+(a), \pi(s_{H_w}^+(a))) &= +(1-\gamma)a^T w + \gamma Q(a,a) \\
&= (1-\gamma)a^T w + \gamma \phi(a,a)^T w \\
&= (1-\gamma)a^T w + \gamma a^T w \\
&= a^T w = \phi(s,a)^T w = Q(s,a),
\end{aligned}
$$

which satisfies the Bellman evaluation equation.

For all cases, $Q$ satisfies the Bellman evaluation equations, so it is the fixed point of the Bellman evaluation operator. In particular, it is the action-value of the policy $\pi$ on $M_{H_w,+}$. The argument is identical for $M_{w,-}$ with $Q(s,a) = -\phi(s,a)^T w$.

$\square$

### F.4 PROOF OF THEOREM 4.5

Consider the MDP class described in Appendix F.1 and Appendix F.2. First, we know from Lemma F.1 that all instances of the BPI and PE problem characterised by the MDP class $\mathcal{M}$ and target policies $\Pi$ satisfy Assumption 4.2 ($Q^\pi$ is realizable for every $\pi$) with the feature map $\phi(\cdot,\cdot)$ defined in Appendix F.1. The proof follows the same reasoning as in Appendix E.4. We consider policy-free queries.

We specify the instance with MDP $M \in \mathcal{M}$ according to the queries selected by the learner. Fix $K > 0$. Recall that $n_k$ is the number of queries made by the learner at round $k$ and $n = \sum_{k=1}^K n_k$ is the total number of queries. Let $A_k$ be the set of (policy-free) actions queried by the learner at round k (i.e. $|A_k| = n_k$). Set $\bar{A}_k = \bigcup_{i=1}^k A_k$ to be the set of all actions queried up to round $k$.

The learner is sample-efficient so $n$ is polynomial in $d$. Specifically, there exists some constant $\alpha > 0$ and some integer $T$ such that $n \leqslant \alpha d^T$. Consider $N \in \mathbb{N}$ s.t. $2^N \leqslant d < 2^{N+1}$ and set $d_+ = 2^N$. Note that $d_+ \geqslant d/2$. Fix $W = \exp(\frac{1}{8}g(\gamma)d_+^{1/4^K})$. If $n \leqslant W$, we can show the learner cannot solve the PE or BPI problems. Consider both cases:

#### F.4.1 CASE 1

Suppose $W < n$, then following the same steps as in E.4.1,

$$W < n \implies K \geqslant c' \log\log d \qquad \text{for a constant } c' > 0 \text{ and } d \text{ sufficiently large.} \tag{4}$$

#### F.4.2 CASE 2

$n \leqslant W \implies n_k \leqslant \exp(\frac{1}{8}g(\gamma)d_+^{1/4^k})$ for $k = 1, ..., K$. Using the same steps as in E.4.2 in the proof of Theorem 4.4 but with $\pi_{H_w}(s^+(a))$ replaced by $s_{H_w}^+(a)$, we can inductively define the sequence of nested subspaces $B_K \subset ... \subset B_1$ characterising $w$ ($\in B_K$) and used in the successor state function $s_M^+$ s.t. for $k = 1, ..., K$, $\dim B_k = 2^{\lceil N/4^k \rceil} \geqslant 2^8$ and

$$\forall a \in \bar{A}_k, \qquad a \notin \overset{\triangle}{\mathcal{C}}_\gamma(B_k).$$

We now fully specify $M$. The complete set of queried actions by the learner is $\bar{A}_K$, and

$$\forall a \in \bar{A}_K, \qquad a \notin \overset{\triangle}{\mathcal{C}}_\gamma(B_K).$$

Since $\dim B_K \geqslant 2^8 > 0$, we can pick some $w \in B_K \cap \partial \mathcal{B}$ and let $H_w = \{B_1, ..., B_K, B_{K+1}\}$ with $B_{K+1} = \langle w \rangle$. From the proof of the existence of nested spaces, we can consider the BPI and PE problem instances with MDPs $M_{H_w,+}$ and $M_{H_w,-}$. The transition function is the same on $M_{H_w,+}$

and $M_{H_w,-}$. Since $\forall a \in \bar{A}_K$, $a \notin \mathcal{C}_\gamma(w) \cup \mathcal{C}_\gamma(-w)$ because $a \notin \overset{\triangle}{\mathcal{C}}_\gamma(B_K)$, the reward function for any queried action $a \in \bar{A}_K$ is 0. Thus, the learner cannot distinguish $M_{H_w,+}$ between $M_{H_w,-}$ from the submitted queries.

For PE, the learner has to produce an estimate of $Q^\pi(\bar{s}, \cdot)$. But

$$Q^\pi(\bar{s}, w) = \phi(\bar{s}, w)^T w = w^T w = 1 \qquad \text{on } M_{H_w,+}.$$
$$Q^\pi(\bar{s}, w) = -\phi(\bar{s}, w)^T w = -w^T w = -1 \quad \text{on } M_{H_w,-}.$$

If the learner predicts a positive value for $Q^\pi(\bar{s}, w)$, it will incur an error greater than 1 on $M_{H_w,-}$ and similarly for $M_{H_w,+}$ if it predicts a negative value. Even if it randomizes between both, with probability at least $1/2$ it will incur an error of at least 1 on one of the MDPs.

Similarly for BPI, the learner has to produce a near-optimal policy in the starting state $\bar{s}$. But

$$V^\star_{M_{H_w,+}}(\bar{s}) = Q^\star_{M_{H_w,+}}(\bar{s}, w) = \phi(\bar{s}, w)^T w = w^T w = 1.$$
$$V^\star_{M_{H_w,-}}(\bar{s}) = Q^\star_{M_{H_w,-}}(\bar{s}, w) = -\phi(\bar{s}, w)^T w = -w^T w = -1.$$

If the learner outputs a policy taking action $a$ s.t. $a^T w > 0$, it will incur an error greater than 1 on $M_{H_w,-}$ and similarly for $M_{H_w,+}$ if it produces an action $a$ s.t. $a^T w \leq 0$. Even if it randomizes between both, with probability at least $1/2$ it will incur an error of at least 1 on one of the MDPs.

Therefore, to be more than $(1, 1/2)$-sound, we must either be in Case 1 or be in Case 2 and satisfy condition (4). In either case, we have the condition

$$K > c \log \log d,$$

for some constant $c > 0$ and $d$ sufficiently large, which gives $K = \Omega(\log \log d)$, showing the result. $\qquad\square$

## G    PROOFS FOR FULLY-ADAPTIVE SETTING

### G.1    PROOF OF THEOREM B.2

Fix an unknown MDP $M$. Consider a learning algorithm with the following procedure:

**Step 1:** The learner selects an arbitrary query $(s_1, a_1)$ s.t. $\|\phi(s_1, a_1)\|_2 = 1$ (possible by Assumption B.1). The learner receives from the environment the reward $r_M(s_1, a_1)$, the transition function $p_M(\cdot|s_1, a_1)$ and evaluations of the target policy $\pi_M$ for all states in the support of the transition function $p_M(\cdot|s_1, a_1)$.

**Step $k \leqslant d$:** For $i < k$, let $(s_i, a_i)$ be the query at round $i$. Define $v_i = \phi(s_i, a_i) - \gamma \mathbb{E}_{s' \sim p_M(\cdot|s_i, a_i)}[\phi(s', \pi_M(s'))]$. Select the query $(s_k, a_k)$ s.t

$$\phi(s_k, a_k) \in \langle v_1, ..., v_{k-1} \rangle^\perp \quad \text{and} \quad \|\phi(s_k, a_k)\|_2 = 1.$$

The feedback to the learner up to round $k$ means the learner has access to $v_1, ..., v_{k-1}$. This together with Assumption B.1 ensures the query-choice is possible.

Denote $v_k = \phi(s_k, a_k) - \gamma \mathbb{E}_{s' \sim p_M(\cdot|s_k, a_k)}[\phi(s', \pi_M(s'))]$.

**Claim:** $v_k \notin \langle v_1, ..., v_{k-1} \rangle$.

*Proof.* Suppose the claim is not true, then $v_k \in \langle v_1, ..., v_{k-1} \rangle$ and $v_k^T \phi(s_k, a_k) = 0$ since $\phi(s_k, a_k) \in \langle v_1, ..., v_{k-1} \rangle^\perp$. Using this,

$$
\begin{aligned}
\|\mathbb{E}_{s' \sim p_M(\cdot|s_k, a_k)}[\phi(s', \pi_M(s'))]\|_2^2 &= \frac{1}{\gamma^2}(v_k - \phi(s_k, a_k))^T (v_k - \phi(s_k, a_k)) \\
&= \frac{1}{\gamma^2}(v_k^T v_k - 2 v_k^T \phi(s_k, a_k) + \phi(s_k, a_k)^T \phi(s_k, a_k)) \\
&= \frac{1}{\gamma^2}(\|v_k\|_2^2 + 1) \\
&\geqslant \frac{1}{\gamma^2} > 1,
\end{aligned}
$$

which is not possible since $\phi(s, a) \in \mathcal{B}$ for all state-actions pairs $(s, a)$, and by Jensen's inequality

$$\|\mathbb{E}_{s' \sim p_M(\cdot|s_k, a_k)}[\phi(s', \pi_M(s'))]\|_2 \leqslant \mathbb{E}_{s' \sim p_M(\cdot|s_k, a_k)}[\|\phi(s', \pi_M(s'))\|_2] \leqslant 1.$$

This proves the claim. $\square$

The claim implies that $\{v_i\}_{i=1}^k$ is a linearly independent set of vectors. To see why this is the case, suppose it is not true:

$$\exists (\alpha_i)_{i=1}^k \quad \text{s.t. one of them is non-zero and} \quad \sum_{i=1}^k \alpha_i v_i = 0.$$

Let $j$ be the largest index s.t. $\alpha_j \neq 0$, then

$$v_j = \frac{1}{\alpha_j}\left(\sum_{i<j} \alpha_i v_i\right) \in \langle v_1, ..., v_{j-1} \rangle,$$

which contradicts the claim. So $\{v_i\}_{i=1}^k$ is a linearly independent set of vectors.

Under realizability, we must have $Q_M^{\pi_M}(s_i, a_i) = \phi(s_i, a_i)^T \theta_M^{\pi_M}$ for some $\theta_M^{\pi_M}$ and since it is the fixed point of the Bellman evaluation operator we also have

$$Q_M^{\pi_M}(s_i, a_i) = r(s_i, a_i) + \gamma \mathbb{E}_{s' \sim p_M(\cdot|s_i, a_i)}[Q_M^{\pi_M}(s', \pi_M(s'))].$$

Let $\Phi = \begin{bmatrix} \phi(s_1, a_1)^T \\ ... \\ \phi(s_d, a_d)^T \end{bmatrix}$, $r = \begin{bmatrix} r_M(s_1, a_1) \\ ... \\ r_M(s_d, a_d) \end{bmatrix}$ and $\Phi^+ = \begin{bmatrix} \mathbb{E}_{s' \sim p_M(\cdot|s_1, a_1)}[\phi(s', \pi_M(s'))^T] \\ ... \\ \mathbb{E}_{s' \sim p_M(\cdot|s_d, a_d)}[\phi(s', \pi_M(s'))^T] \end{bmatrix}$.

Combining the realizability assumption with the Bellman fixed point equation with the above notation we have:

$$\Phi \theta_M^{\pi_M} = r + \gamma \Phi^+ \theta_M^{\pi_M} \iff \left(\Phi - \gamma \Phi^+\right) \theta_M^{\pi_M} = r.$$

Noticing that $v_i$ is the $i$th row of $\Phi - \gamma \Phi^+$ and using that they are all linearly independent, $\Phi - \gamma \Phi^+$ is a square full rank matrix and is thus invertible, giving the unique solution of the policy evaluation problem $\theta_M^{\pi_M}$ in terms of quantities known to the learner at the $d$-th round. $\qquad \square$

### G.2 PROOF OF THEOREM B.3

#### G.2.1 MDP CLASS

We consider the same MDP class as for Theorem F - see Appendix F.1.

#### G.2.2 INSTANCE OF THE CLASS

Consider a sequence of $d$ strictly nested subspaces of $\mathbb{R}^d$:

$$B_{d-1} \subset B_{d-1} \subset ... \subset B_2 \subset B_1 \subset B_0 = \mathbb{R}^d,$$

s.t. $\dim B_k = d - k$. Since $\dim B_{d-1} = 1$, there is some $w \in B_K \cap \partial \mathcal{B}$ s.t. $B_{d-1} = \langle w \rangle$. Let $H_w = \{B_1, ..., B_{d-1}\}$ denote the set of nested subspaces. Every MDP $M \in \mathcal{M}$ is fully characterised by the sequence of subspaces $H_w$ and a sign $+$ or $-$. Hence, they are denoted by $M_{H_w,+}$ or $M_{H_w,-}$.

**Reward function:** The reward function only depends on the vector $w$ and the sign $\pm$ but we denote them with the same subscript as the MDP. Specifically it is defined

$$\text{on } M_{H_w,+} : \qquad r_{H_w,+}(s,a) = \begin{cases} 0, & \text{if } a \notin \mathcal{C}_\gamma(w) \cup \mathcal{C}_\gamma(-w). \\ +(1-\gamma)a^T w, & \text{otherwise.} \end{cases}$$

$$\text{on } M_{H_w,-} : \qquad r_{H_w,-}(s,a) = \begin{cases} 0, & \text{if } a \notin \mathcal{C}_\gamma(w) \cup \mathcal{C}_\gamma(-w). \\ -(1-\gamma)a^T w, & \text{otherwise.} \end{cases}$$

See Appendix C for the definition of hyper-spherical caps $\mathcal{C}_\gamma(w)$.

**Transition Function:** The transition function for an MDP $M \in \mathcal{M}$ depends on the sequence of nested subspaces $H_w$ but not on the sign $\pm$. The transition function is therefore the same for $M_{H_w,+}$ and $M_{H_w,-}$. If $A$ is a subspace of $\mathbb{R}^d$, let $\text{proj}_A(x)$ denote the orthogonal projection of $x$ onto $A$.

See Appendix C for the definition of hyper-spherical caps $\mathcal{C}_\gamma(w)$ and sectors $\overset{\triangle}{\mathcal{C}}_\gamma(H)$. The successor state of a state-action pair $(s,a)$ is deterministic and only depends on the chosen action (and not the current state), so we will denote the unique successor state when taking action $a$ by $s_{H_w}^+(a) = a$, which is defined as:

$$s_{H_w}^+(a) = \begin{cases} \frac{1}{\gamma}\text{proj}_{B_{k+1}}(a), & \text{if } a \in \overset{\triangle}{\mathcal{C}}_\gamma(B_k) \backslash \overset{\triangle}{\mathcal{C}}_\gamma(B_{k+1}) \text{ for } k = 0, ..., d-2. \\ \frac{1}{\gamma}\text{proj}_{B_{d-1}}(a), & \text{if } a \in \overset{\triangle}{\mathcal{C}}_\gamma(B_{K+1}) \backslash (\mathcal{C}_\gamma(w) \cup \mathcal{C}_\gamma(-w)). \\ a, & \text{if } a \in \mathcal{C}_\gamma(w) \cup \mathcal{C}_\gamma(-w). \end{cases}$$

This is defined in the same way as in the proof of Theorem F - we refer the reader to Appendix F.1 for an explanation of why this definition is well defined.

Crucially actions queried in $\overset{\triangle}{\mathcal{C}}_\gamma(B_k) \backslash \overset{\triangle}{\mathcal{C}}_\gamma(B_{k+1})$ for $k \leqslant d-2$ does not reveal $w$. Without the knowledge of $w$, the reward function is unknown and even with the knowledge of $w$ the reward function is not fully identified, in which case $M_{H_w,+}$ and $M_{H_w,-}$ cannot be distinguished.

#### G.2.3 PROOF OF THEOREM B.3

Consider the MDP class described in Appendix G.2.1 and Appendix G.2.2. First, we know from Lemma F.1 that all instances of the BPI and PE problem characterised by the MDP class $\mathcal{M}$ and

target policies $\Pi$ satisfy Assumption 4.2 ($Q^\pi$ is realizable for every $\pi$) with the feature map $\phi(\cdot, \cdot)$ defined in Appendix F.1.

**Defining the subspaces in** $H_w$**:** Denote the first $d-1$ queries chosen by the learner by $(s_1, a_1), ..., (s_{d-1}, a_{d-1})$. In the feature space, these are $a_1, ..., a_{d-1}$. Let $B_k$ be the orthogonal complement of $\langle a_1, ..., a_k \rangle$, i.e.

$$B_k = \langle a_1, ..., a_k \rangle^\perp = \left\{ x \in \mathcal{B} : x^T a = 0 \quad \forall a \in \langle a_1, ..., a_k \rangle \right\},$$

and $w \in B_{d-1}$. The definition of $B_k$ is well defined since the feedback of $\{a_1, ..., a_{k-1}\}$ only depends on $B_1, ..., B_{k-1}$: for $i = 1, ..., k-1$, $s_{H_w}^+(a_i) = \frac{1}{\gamma}\text{proj}_{B_i}(a_i), r_{w,\pm}(s, a_i) = 0$. The learner only observes the projection of actions in $\{a_1, ..., a_{k-1}\}$ into $B_1, B_2, ..., B_{k-1}$. In particular, $B_k$ can be fixed as any subspace nested in $B_{k-1}$ once the learner has chosen the action-query $a_k$ at round.

In Appendix G.2.2, we fixed $\dim B_k = d - k$. If $a_1, ..., a_k$ are not linearly independent, the dimension of $B_k$ may be greater than $d - k$. In this case, we just restrict $B_k$ arbitrarily such that the sequence remains nested. In particular, we have $\dim B_{d-1} = 1$ and $B_{d-1} = \langle w \rangle$ for some $w \in \partial\mathcal{B}$. Let $H_w = \{B_1, ..., B_{d-1}\}$. Consider the BPI and PE problem instances with MDPs $M_{H_w,+}$ and $M_{H_w,-}$. The transition function is the same on $M_{H_w,+}$ and $M_{H_w,-}$.

By construction $a_i \notin \overset{\triangle}{\mathcal{C}}_\gamma(B_{d-1})$ for all $i \leq d-1$. In particular, $a_i \notin \mathcal{C}_\gamma(w) \bigcup \mathcal{C}_\gamma(-w)$ for $i \leq d-1$ and the reward function for any queried action is $0$. Thus, the learner cannot distinguish $M_{H_w,+}$ between $M_{H_w,-}$ from the submitted queries.

For PE, the learner has to produce an estimate of $Q^\pi(\bar{s}, \cdot)$. But

$$Q^\pi(\bar{s}, w) = \phi(\bar{s}, w)^T w = w^T w = 1 \qquad \text{on } M_{H_w,+}.$$
$$Q^\pi(\bar{s}, w) = -\phi(\bar{s}, w)^T w = -w^T w = -1 \quad \text{on } M_{H_w,-}.$$

If the learner predicts a positive value for $Q^\pi(\bar{s}, w)$, it will incur an error greater than 1 on $M_{H_w,-}$ and similarly for $M_{H_w,+}$ if it predicts a negative value. Even if it randomizes between both, with probability at least $1/2$ it will incur an error of at least 1 on one of the MDPs.

Similarly for BPI, the learner has to produce a near-optimal policy in the starting state $\bar{s}$. But

$$V^\star_{M_{H_w,+}}(\bar{s}) = Q^\star_{M_{H_w,+}}(\bar{s}, w) = \phi(\bar{s}, w)^T w = w^T w = 1.$$
$$V^\star_{M_{H_w,-}}(\bar{s}) = Q^\star_{M_{H_w,-}}(\bar{s}, w) = -\phi(\bar{s}, w)^T w = w^{-T} w = -1.$$

If the learner outputs a policy taking action $a$ s.t. $a^T w > 0$, it will incur an error greater than 1 on $M_{H_w,-}$ and similarly for $M_{H_w,+}$ if it produces an action $a$ s.t. $a^T w \leq 0$. Even if it randomizes between both, with probability at least $1/2$ it will incur an error of at least 1 on one of the MDPs.

Therefore, the learner can be at most $(1, 1/2)$-sound for both PE and BPI problems with $n = K \leq d - 1$ queries. To be more than $(1, 1/2)$-sound, $n = K \geq d$ queries are required. $\qquad\square$

