# OpenReview forum: "Sample-Efficiency in Multi-Batch Reinforcement Learning: The Need for Dimension-Dependent Adaptivity"
_ICLR.cc/2024/Conference — ICLR 2024 poster_

### Official Review · Reviewer_ycxM · 2023-10-31

**Soundness:** 4 excellent
**Presentation:** 3 good
**Contribution:** 4 excellent
**Rating:** 8
**Confidence:** 3

**Summary:**

This paper studies how many rounds $K$ of adaptive queries $n$ are required to learn MDPs with linear function approximation in $d$ dimensions, with 1 round corresponding to offline/batch setting, and $K = n$ corresponding to an online (or more accurately, generative access) setting.

They provide a lower bound and show that any algorithm which makes poly(d) total queries must use at least $\Omega(\log \log d)$ rounds. In particular, the number of rounds for sample efficient algorithms must depend on the dimension. The proof leverages some recent results in subspace covering (Soleymani & Mahdavifar, '21).

**Strengths:**

- The paper is well written and easy to digest. The results / contributions are easy to pick out, and the proof sketch clearly captures the high-level intuition. The paper also makes clear the relationship to prior / related work.
- The paper asks an important and interesting question about adaptivity, and the contribution is solid.

**Weaknesses:**

- In the later version, it would be nice to have diagrams illustrating the hyperspherical cap / sector as well as the policy class considered in the lower bound.
- I still lack some intuition for how the policy class is constructed, and why it is hard to learn.
- Perhaps some discussion on whether the lower bound can be improved, i.e., what sort of technical results are needed in order for the current construction to strengthen the lower bound.

Minor typo: on page 21, the definition of $W$ needs to have \exp. Also, where is $g(\gamma)$ defined? I see it used in Lemma C.5, but I wasn't sure if it was properly defined anywhere.

**Questions:**

1. It wasn't obvious to me why finite horizon linear MDPs can be solved with number of deployments independent of $d$. More generally, is the gap really between finite horizon vs infinite horizon, or between linear MDPs and your $Q^\pi$ realizability assumptions (4.1 and 4.2)?
2. Do you expect to be able to improve the lower bound significantly? Perhaps, to log d or even some polynomial in d?

---

> ### Author Response · Authors · 2023-11-18
> **Response to reviewer ycxM**
>
> We thank reviewer ycxM for their valuable comments and feedback.
>
> $~$
> ## C.1 Diagrams and Intuition for Policy Class
>
> We have uploaded a new pdf version of the paper that now contains a section with diagrams illustrating the construction of the hard MDP instance (Appendix D.5, pages 28-29) as well as a diagram of hyper-spherical sectors in Appendix B. We hope this provides some clarity and better intuition.
>
>
> $~$
> ## C.2 Improving the Lower-Bound
>
> In the final version, with the additional allowed page, we will include this discussion on the improvement of the lower-bound:
>
> To improve the lower-bound with the current construction, we would require the existence of a subspace packing with minimal chordal distance between subspaces $d_{\min} (\mathcal{C}) \geq \sqrt{d - g(\gamma)^2}$ (where $g(\gamma) = 2\gamma^2 - 1$) whose size is exponential in $d$ and the dimension of the subspaces is some constant fraction of $d$ (instead of $d^{1/4}$ as we have currently, this would be the crucial difference), say $c d$ for some $c < 1$. After $k$ rounds, information would be missing along a subspace of dimension $c^k d$, (instead of $d^{1/4^k}$) from which we could get a $\log d$ lower-bound. As far as we are aware, such a result is not available in the subspace packing literature, nor is a subspace covering result that would rule out the possibility of such a packing. We highlight that our MDP construction can be combined with any other subspace packing result, paving the way for improved lower-bounds should new subspace packing procedures be developed.
>
>
>
>
>
>
>
> $~$
> ## C.3 Typos
>
> Thank you for pointing this typo out! As for $g(\gamma)$, it is defined in the statement of Lemma C.5 but in fact we use it before this (e.g.\ statement of Lemma C.4) - **we will properly define it earlier on in Appendix C**.
>
> $~$
> ## C.4 Finite Horizon
>
> **Finite-horizon linear MDPs can even be solved in the offline setting** ($K=1$) with a total of $d \cdot H$ queries (where $Q^\pi_t$ realizability is assumed at each step $t \in \{1, ..., H\}$). This can be seen by at each step $t$ selecting $d$ queries whose features form a basis and then running a backward induction (See Appendix D.2 of Zanette (2021) for details).
>
> Therefore, under our $Q^\pi$ realizability assumption, **there is a gap between infinite horizon and finite horizon MDPs**. There may also exist a gap between linear MDPs and our $Q^\pi$ realizability assumption, since the former is a stronger assumption.
>
> $~$
> ## C.5 Improving the Lower-Bound
>
> See C.2 for a discussion of what is required to improve the lower-bound.
>
> As we have stressed in the conclusion section, it is unclear if the $\log \log d$ dependence on $d$ is tight, which we leave as an open question. We add here that the subspace packing result we use currently is not tailored to our setting in the sense that we associate a learner's query with an entire subspace and use that all vectors in that subspace are far enough in inner product from vectors of any other subspace in the packing. But we only need this for the query not the entire subspace associated to the query. Therefore there is room for improvement with some more specific subspace packing reasoning but it remains unclear if this would allow an improvement to $\log d$ or higher.

---

> > ### Comment · Reviewer_ycxM · 2023-11-22
> > **Reviewer Acknowledgement**
> >
> > Thank you for your detailed response. I have no further questions at this point.

---

### Official Review · Reviewer_XsoY · 2023-10-31

**Soundness:** 3 good
**Presentation:** 3 good
**Contribution:** 3 good
**Rating:** 5
**Confidence:** 3

**Summary:**

This paper studies sample efficiency in offline reinforcement learning with linear function approximation. In detail, it shows a sample efficiency separation gap between the adaptive setting (where the data is obtained in multiple batches, and the data obtained later is selected based on the information of the reward or transition induced by previous data) and the non-adaptive setting (where the data can only be obtained based on the linear feature itself, not on the unknown environment). The authors claimed that when the number of batches is less than $\log\log d$ where $d$ is the dimension of the linear feature, for any reasonable learner, there always exists a hard environment can not be learned by it efficiently.

**Strengths:**

The presentation of this paper is clear.

The theoretical proof is technically sound.

**Weaknesses:**

The importance of the proposed problem setting is unclear to me. Overall the problem setting tries to suggest that if number of adaptivity is too small (less than $\log\log d$), then the $d$ dimension feature space can not be fully spanned by given features induced by the data obtained so far, and the solution $\theta$ which admits the value function (either optimal value function or value function associated with the target policy need to be evaluated) is not unique, which gives a large value gap when we consider the worst state-action pair we want to evaluate. However, it seems that such a separation gap becomes not important if we admit
1.  A regret guarantee instead of a worst-case evaluation error guarantee (as defined in Def 3.1). For the regret guarantee, since we only care about the value gap over \bf{existing} state-action pairs (due to the definition of regret) rather than the state-action pair that achieves the maximum which we may never faced before (again, due to Def 3.1), the nonuniqueness of $\theta$ will not affect the regret as it affects the policy evaluation error. Is it true that such a lower bound does not hold if we slightly revise the definition of Def 3.1 from $P(\sup_{a \in \mathcal{A}}(\cdots))$ to $P(\mathbb{E}_{a \sim \mu}(\cdots))$ where $\mu$ is some distribution over $\mathcal{A}$?
2. Any limitation for the value gap (e.g., the sub optimality gap defined in [1]) or the size of action space $|\mathcal{A}$, or some coverage assumption which assumes that the obtained data can cover the target policy. To me it seems that if any of the above assumptions hold, then the separation gap does not hold either. Meanwhile, since these assumptions are also considered as mild assumptions (which hold in many settings), the proposed result in this paper is less interesting.

[1] He, Jiafan, Dongruo Zhou, and Quanquan Gu. "Logarithmic regret for reinforcement learning with linear function approximation." International Conference on Machine Learning. PMLR, 2021.

Meanwhile, the problem setting and the hard instance construction are also very similar to that in Zanette 2021, which dwarfs the technique contribution in this work. More comparison between the technique difficulty faced in this work and that in Zanette 2021 are welcomed.

**Questions:**

See Weaknesses section.

---

> ### Author Response · Authors · 2023-11-18
> **Response to reviewer XsoY**
>
> We thank reviewer XsoY for their valuable comments and feedback.
>
> $~$
> ## B.1 Weaknesses of Proposed Problem Setting
>
> The mentioned weakness on considering a worst-case evaluation error (instead of a regret guarantee) (point 1.) **does not apply to the best policy identification** (BPI) problem for which there is no worst-case evaluation (Definition 3.2). For the policy evaluation (PE) problem, it is true that the lower bound no longer holds for the proposed definition of Definition 3.1. Though this is not the setting we study and our definition (and results) remains relevant **if the interest is a worst-case policy evaluation error**.
>
> As for point 2. on the additional assumptions not covered by our results, it is also **common to have an infinite action space and no additional assumption on the sub-optimality gaps**. Additionally, while it may be necessary to have coverage assumptions in an offline setting where the learner has no control over the data collection, this **assumption need not be made when data is collected adaptively**, as in our setting, since the learner is selecting the state-action pairs to query. For PE, in our second result (Theorem 4.5), the target policy can be fully revealed before collecting any data so the learner can choose the data with knowledge of the target policy (and indeed the hardness is introduced by the worst-case evaluation error). It is an **interesting direction to understand if our results continue to hold under these additional assumptions** or if these assumptions lead to a difference in the sample-efficiency of low-adaptivity RL. We leave this as future-work and will include a discussion about these assumptions in the final version.
>
> We note that **this setting has been considered in prior work** (in particular Zanette (2021)) to establish some important results from the existing literature.
>
>
>
>
>
>
> $~$
> ## B.2 Comparison to Zanette (2021)
>
> Please refer to Section A.6 in response to reviewer 7s3L for an additional comparison that we will include in Section 5.
>
> The technical novelty of our work is using (as mentioned by reviewer 7s3L)  **``novel tools such as subspace packing with chordal distance" (Soleymani \& Mahdavifar, 2021) and applying them to our setting** (see Appendix C). These tools have not been applied in an RL setting before as far as we are aware. In particular, the technical challenges include the following points.
> * 1. Relating the subspace packing result with chordal distance to the existence of a subspace $B_k$ that is far (does not intersect the $\gamma$-hyperspherical cones) from all the learner's queries up to round $k$ if there are polynomial in $d$ number of queries (Section C). Information can then be erased in the directions of the subspace $B_k$ whose dimension is large if $k \neq \Omega(\log \log d)$.
>
> The work of Zanette (2021) only requires showing the existence of a $1$-dimensional subspace. A **volumetric argument** can be used in their setting: since the sum of the volumes of an exponential (in $d$) number of hyper-spherical cones is less than the volume of the unit-sphere, there must exist a point in the sphere not covered by these exponentially many cones. This point can be used as the 1-dimensional subspace. However, **we cannot proceed with a similar volumetric argument** because we consider $m$-dimensional subspaces. The space in the sphere left vacant by the cones of the learner's queries can easily be used to argue the existence of individual points not covered by the cones, but not the existence of subspaces of multiple dimensions. Instead, as mentioned above, we adapt tools from the theory of subspace packing.
>
> * 2. Using the existence of these subspaces to extend the hard instance construction of Zanette (2021) to our setting over multiple adaptive rounds, while preserving the realizability assumptions. The extension over multiple rounds requires a careful treatment of the dependence of the queries on observed feedback (see Appendix D.4.2/E.4.2) to establish the existence of these subspaces, which does not arise when considering a single round as in Zanette (2021).
>
> We have uploaded a new pdf version of the paper that now contains a section with diagrams illustrating the construction of the hard MDP instance (Appendix D.5, pages 28-29). We hope this provides some more clarity on the technical differences with Zanette (2021).
>
> Aside from the technical novelties outlined above, we would like to stress the novelty of our main result, which is to show the need for dimension-dependent adaptivity in RL under linear function approximation.

---

### Official Review · Reviewer_7s3L · 2023-11-02

**Soundness:** 4 excellent
**Presentation:** 3 good
**Contribution:** 4 excellent
**Rating:** 6
**Confidence:** 2

**Summary:**

This paper studies multi-batch reinforcement learning, where the learner is allowed to query the oracle multiple times and adaptively adjust the policies. The goal is to evaluate a policy/identify the best policy. For this problem, the authors show that to achieve sample efficiency, the number of batches K has to grow at least on the order of loglog d.

**Strengths:**

Significance of the contribution: This paper provides a novel lower bound for the number of batches needed to achieve sample efficiency for the multi-batch reinforcement learning problem, which is an excellent contribution. This lower bound shows that this number increases with loglogd, so a constant number is not enough, which is a very meaningful result. It also indicates that, adaptivity is necessary for achieving sample efficiency, and the dependence on d is not negligible. Finally, this result is also inspiring and leaves many questions open, such as whether this lower bound is tight, or how to obtain matching lower bound.

Novelty of the method: The lower bound is constructed based on extending the proof techniques in Zanette (2021), but with novel tools such as subspace packing with chordal distance.

Presentation: this paper is in general well-written and easy to read. I in particular appreciate the proof sketch section, which helps the reader understand the general proof idea.

**Weaknesses:**

As mentioned before, the paper provides a lower bound for K, but it is not clear whether it is tight.  loglogd is also sometime  considered as a constant is some works, since lnln 10^9 is only approximately 3.

I have some questions on some details of the paper, which is listed in the next section.

**Questions:**

Questions:

Section 4: “we assume gamma\geq sqrt{3/4}”. Can the authors provide some justification/context for this assumption? It is purely an assumption to make the analysis work?


The proof of Theorem 4.4 (Appendix D) and that of Theorem 4.5 (Appendix E) seem very similar, in particular, some subsections, such as D.4.1 and E.4.1, seem to be exactly the same. So I was wondering what is the relationship between the two proof? Is one proof just an simple extension of another?

Either way, I suggest the authors avoid copy-past proof so that the proof can be shorter.


For, Theorems 4.4 and 4.5, does the conclusion only holds for a particular pair of epsilon, delta, e.g., 1, 1/2, or the a more general conclusion can be easily obtained?

I don’t quite understand the second paragraph of Section 4, partly because it is a short paragraph with lots of information. In particular, the authors mentioned the established a *matching* lower bound for some problem while the upper bound is d. How is this lower bound relates to the main result?

At the beginning of Section 5, the authors mentioned that their proof extends that in Zanette (2021). However this paper is not mentioned at all in the rest parts of this Section. It would be great if the authors can compare the idea here with Zanette (2021) in the proof sketch section.

---

> ### Author Response · Authors · 2023-11-18
> **Response to reviewer 7s3L**
>
> We thank reviewer 7s3L for their valuable comments and feedback.
> $~$
> ## A.1 Weaknesses
>
>
> The significance of the presented result is in the **existence of a dependence on the dimension $d$** for the amount of adaptivity $K$. From a theoretical perspective, this represents an important difference to $K$ being constant with respect to $d$. Our results provide an **important step in studying the limits of RL under low-adaptivity by showing a dependence on dimension is unavoidable** but much is left to understand. In particular, as we stress in the conclusion, it remains unclear if the $\log \log d$ dependence on $d$ is tight, which we leave as an open question.
>
> $~$
> ## A.2 Assumption that $\gamma \geq \sqrt{3/4}$
>
> It is an assumption to **make the analysis work**. In the analysis, we show the existence of subspaces that are at least some chordal distance away from each other and the distance depends on $g(\gamma) = 2\gamma^2 - 1$. We assume $\gamma \geq \sqrt{3/4}$ to ensure $g(\gamma)$ is not too small, which allows us to get the right dependence on the dimension $d$. Given that the discount factor $\gamma$ in discounted MDPs is considered to be close to 1, this **assumption is not restrictive**.
>
> $~$
> ## A.3 Relationship between Theorems 4.4 and 4.5
>
> The proofs of both Theorems are based on the same idea of the existence of a sequence of subspaces but the way the constructions of the MDP classes incorporate this sequence of subspaces is different. Therefore once the proofs have been reduced to using the existence of this sequence of subspaces, they are essentially the same. In the final version, **we will better unify these sections** to avoid repetition.
>
> $~$
> ## A.4 Choice of epsilon, delta for Theorems 4.4 and 4.5
>
> The proof of both Theorems conclude with the existence of two MDPs which are indistinguishable given the queries made by the learner. One of them has value $+1$ and the other $-1$ (optimal state-value for best policy identification (BPI) and action-value of the target policy for policy evaluation (PE)) in the initial state.
>
> If the learner aims to minimise $\delta$, it can randomise equally between $+1$ and $-1$ and have an error of $2$ with probability $1/2$. If the learner aims to minimise $\varepsilon$, it can deterministically output $0$ and have an error $\varepsilon$ of $1$ with probability $1$.
>
> The values of $\varepsilon = 1$, $\delta = 1/2$ from Theorems 4.4 and 4.5 comes from taking the learner's best-case in the combination of these. However, we obtain the more general conclusion that any learner must be between $(2,1/2)$-soundeness and $(1,1)$-soundeness. The $(1,1)$-soundeness covers any value of $\varepsilon < 1$ and the $(2,1/2)$-soundeness with $\varepsilon = 2$ is the largest error value since our results hold for MDPs whose values are within $[-1,1]$ (we did not explicitly point this out but will in the final version). **Therefore our results cover all relevant $\varepsilon$ values (for which there is a trade-off with $\delta$)**.
>
> $~$
> ## A.5 Fully-Adaptive Setting
>
> "I don’t quite understand the second paragraph of Section 4" - we believe you meant Section 4.3 but please correct us if not.
>
> This paragraph is related to the **fully-adaptive** setting where adaptivity constraints are removed and the number of batches $K$ is equal to the number of queried data-points $n$. This is in contrast to our main results (Theorems 4.4 and 4.5), which show $K > \log \log d$ are needed to solve BPI/PE when $n \gg K$ but $n$ is polynomial in $d$. In Appendix A, we show that in the fully-adaptive setting, $n \geq d$ is needed to solve BPI/PE to arbitrary accuracy (and therefore $K\geq d$ since $K = n$). We also show a matching upper bound of $K=n\leq d$ to fully solve a PE problem under an assumption on the action space, hence the matching upper and lower bound.
>
> Therefore the **relation between the lower-bound in the fully-adaptive setting and the main results is that they consider different settings of how $K$ relates to $n$**.
>
> $~$
> ## A.6 Comparison to Zanette (2021) in Proof Sketch
>
> In the final version, with the additional allowed page, we will include this comparison to the work of Zanette (2021) in Section 5:
>
> **Comparison to Zanette (2021):** The work of Zanette (2021) erases information along a 1-dimensional subspace ($X$ is of rank $d-1$). Therefore after having observed one round of feedback, the learner only needs a single query to fully distinguish the MDP and solve the problem. Our constructions erase information along $m$-dimensional subspaces where we attempt to maximise $m$ so that even after having observed feedback revealing this subspace, a polynomial number of queries remains insufficient to distinguish the MDP and solve the problem, more adaptive rounds are needed.

---

### Meta-Review · Area_Chair_qrNf · 2023-12-06

**Metareview:**

This paper investigates the relation between sample-efficiency and adaptivity in reinforcement learning (RL). A sample-efficient algorithm is defined as one that uses a number of queries to the environment polynomial in the problem's dimension. Adaptivity, on the other hand, refers to the frequency of queries and feedback processing for updating the policy. The results in this paper reveal that having adaptivity doesn't automatically ensure sample-efficiency. Notably, the adaptivity-boundary for sample-efficiency doesn't lie between offline RL and adaptive online RL. Instead, it varies across different adaptivity regimes and depends on the problem dimension. This paper has received general support from the reviewers. I concur with their evaluation and recommend acceptance.

**Justification For Why Not Higher Score:**

The audience for this paper may be limited due to its pure theoretical results.

**Justification For Why Not Lower Score:**

This paper has received general support from the reviewers.

---

### Decision · Program_Chairs · 2024-01-16

Accept (poster)